# DG-LLM: Decomposition-based dynamic graph adaptation of large language models for spatiotemporal traffic forecasting

**Sadia Tabassum**[¤], **Naushin Nower**[¤*]

Institute of Information Technology, University of Dhaka, Dhaka, Bangladesh

¤ Current Address: Institute of Information Technology, University of Dhaka, Dhaka, Bangladesh
* naushin@iit.du.ac.bd

## Abstract

Traffic forecasting plays a critical role in the field of urban planning. Yet, existing methods struggle with modeling complicated spatiotemporal dependencies and capturing long-term patterns due to their multiscale nature. In this paper, we present a novel framework named DG-LLM that leverages the advantages of decomposed temporal representations and adaptive spatial connectivity to model spatiotemporal dependencies. In this framework, traffic signals are decomposed into intrinsic modes, and dynamic graphs are learned for each mode to represent the spatial dependencies. These representations are then incorporated with pre-trained Large Language Models for effective long-range temporal dependency modeling. We conducted comprehensive experiments across six real-world traffic datasets spanning urban mobility systems and highway traffic networks and evaluated short- and long-term forecasting. Experimental results demonstrate that our framework provides significant improvements over state-of-the-art approaches, including benchmark graph- and LLM-based spatiotemporal forecasting models, even in long-term forecasting scenarios with severe temporal instability. Our model outperforms other methods by achieving $13 - 19\%$ improvements in MAE and $19 - 25\%$ in RMSE across all six benchmarks compared with baseline approaches. Additional analyses, including ablation studies, robustness to missing data, and zero-shot cross-dataset evaluation, further validate the effectiveness and generalization capability of the proposed framework.

## Introduction

Traffic congestion has become one of the most critical problems faced by cities in recent times, which has a long-lasting effect on the economy, environment, and society. In congested urban cities, traffic congestion affects productivity, consumption of fuel, and greenhouse gas emissions. For instance, in the US alone, the total cost of traffic congestion to commuters is 269 billion annually as of 2024, indicating a 16% increase over the past five years [1]. Urban cities experience severe traffic

**Data availability statement:** All relevant data are available from the GitHub repository at https://github.com/SadiaTabassum1216/DG-LLM.

**Funding:** The author(s) received no specific funding for this work.

**Competing interests:** The authors have declared that no competing interests exist.

congestion daily, resulting in millions of lost working hours and economic costs annually [2]. Moreover, traffic congestion affects the quality of life of individuals living in urban cities by causing stress and reducing time for various activities. All of these issues make traffic forecasting a cornerstone for intelligent traffic management in urban cities.

Traffic predictions can allow transportation authorities to anticipate congestion patterns, help optimize the flow of traffic, and make policies accordingly. This makes traffic prediction an integral part of the Intelligent Transport System (ITS). However, accurate traffic forecasts are difficult to obtain due to the complex nature of traffic systems.

The dynamics of traffic flow are controlled by numerous factors. These factors include spatial dependencies and temporal dependencies, as illustrated in Fig 1. Spatial dependencies illustrate how traffic flow in a particular section of the road influences flow in adjacent sections. Temporal dependencies illustrate the dynamic changes in traffic flow over time, driven by periodicity and trend [3]. Non-stationarity is also inherent in traffic time series, where long-term trends, regular daily and weekly periodicity, and irregular fluctuations caused by accidents, weather conditions, holidays, and special events coexist [4]. These dynamics collectively shape complex spatiotemporal dependencies that must be effectively modeled for accurate traffic forecasting. Moreover, traffic patterns exhibit long-range temporal correlations, where distant historical patterns can also affect future states. Effective forecasting models, therefore, must capture multi-scale temporal behavior while simultaneously modeling evolving spatial interactions across complex road networks.

Conventional statistical models, such as the Historical Average (HA), Auto-Regressive Integrated Moving Average (ARIMA) [5], and Kalman filtering [6] are mostly based on stationary data and do not account for spatial relationships. These models are found to be inadequate for real-world traffic data that is non-linear and non-stationary [7]. The use of deep learning models has also shown better results by directly learning from data. Earlier models, such as Recurrent Neural Networks (RNNs), Long Short-Term Memory (LSTMs), and Gated Recurrent Units (GRUs), are mainly based on temporal relationships. These models are found to be inadequate for long-term forecasting due to gradient vanishing and gradient explosion. The more recent models use Convolutional Neural Networks (CNNs) for spatial relationships, along with RNNs or Temporal Convolutional Networks (TCNs) for temporal relationships. Yet, conventional CNNs are inherently designed for Euclidean, grid-structured data (e.g., images or regular grids) and do not naturally generalize to irregular, non-Euclidean road networks.

The Graph Neural Network (GNN) provides a more intuitive formulation through graph-based modeling of traffic networks, in which nodes are considered as sensors and edges as connections between these sensors. Spatiotemporal GNN models like DCRNN [8] and STGCN [9] have shown potential through the use of graph convolutions with temporal modeling. However, most GNN models use a static adjacency matrix, assuming a fixed spatial relationship. This restricts the adaptation of dynamic traffic conditions and congestion propagation. This necessitates a shift toward

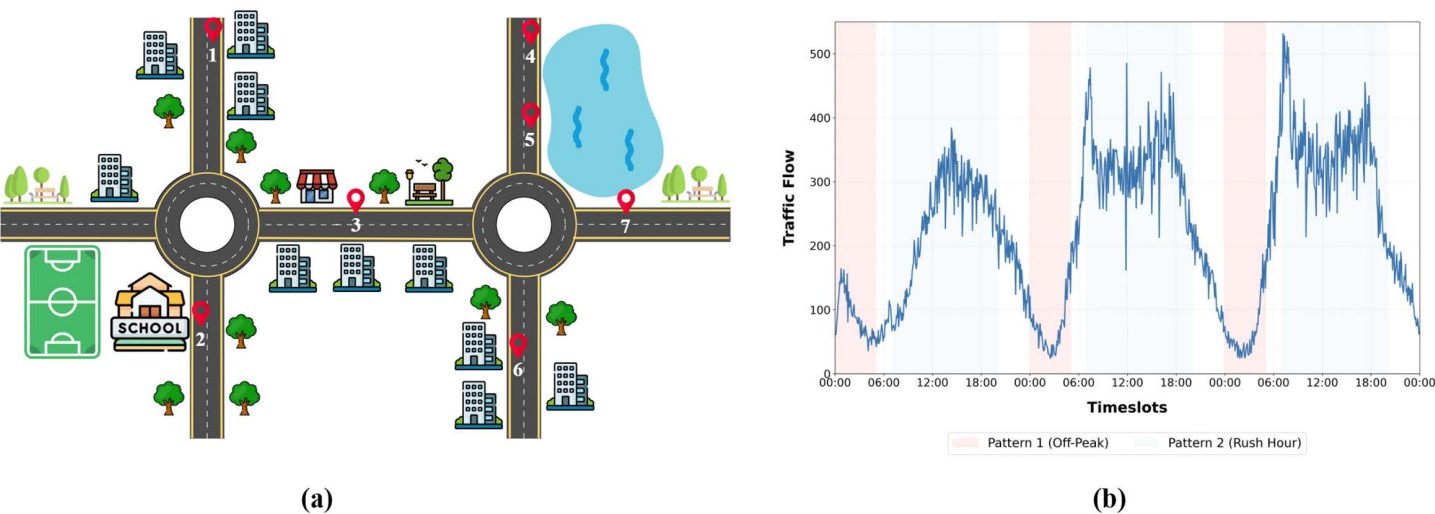

**(a)**                                                                                                           **(b)**

**Fig 1. Spatial and temporal dependency among road sensors.** (a) Sensors 1 & 6 or sensors 4 & 5 show similar patterns, whereas they differ from sensor 2. (b) A daily traffic pattern is observed on three consecutive days.

dynamic graph architectures that can adaptively capture time-varying spatial dependencies and long-range correlations that are otherwise obscured by static topologies.

Recently, Large Language Models (LLMs) based on the Transformer-based architectures have demonstrated remarkable success across tasks beyond language processing, including vision and time-series forecasting [7]. These models are particularly efficient at modeling long-term dependencies via continuous embeddings and demonstrate strong generalization via few-shot and zero-shot learning. LLM architectures are initially designed to work with text data, such as words and tokens. However, these are currently being adapted to cater to the demands of multivariate traffic data. Existing LLM-based time-series modeling techniques mostly overlook spatial information or include it only in the prompt, hence do not apply to traffic data. Recently introduced LLM-based spatiotemporal techniques, such as STLLM [7] and STLLM+ [10], aim to incorporate graph information. Nevertheless, these techniques rely on static graph structures and cannot dynamic, evolving behavior of traffic flow. There is a critical gap in existing techniques, as they cannot analyze the multidimensional, intertwined patterns in traffic data.

To mitigate this, decomposition-based techniques offer several benefits for traffic forecasting by addressing non-stationarity and complexity in traffic signals. Several techniques, such as the Wavelet Transform, Empirical Mode Decomposition (EMD), and Variational Mode Decomposition (VMD), are employed to decompose a complex time-series signal into sub-signals that are easier for forecasting models to learn. Thus, decomposition reduces signal interference, allowing models to focus on more homogeneous subpatterns. In this regard, VMD is more favorable as it produces stable, well-separated intrinsic modes with clear frequency boundaries, which are easier for models to learn [11]. Moreover, different nodes of a traffic network are spatially related at different levels, depending on the mode of decomposition. For example, long-range relationships between distant nodes are captured by low-frequency modes (trends), while relationships between neighboring nodes are captured by high-frequency modes (noise). Therefore, the construction of a dynamic graph must be adaptive to different decomposition modes.

Inspired by these observations, we propose DG-LLM (**D**ecomposition-based Dynamic **G**raph Adaptation of **L**arge **L**anguage **M**odel), a novel spatiotemporal forecasting framework built on a graph-aware and parameter-efficient LLM backbone. The proposed approach captures the intrinsic temporal modes of traffic signals, learns mode-dependent dynamic spatial relationships, and integrates these representations into a pretrained LLM to improve long-range forecasting.

DG-LLM introduces a new paradigm for spatiotemporal forecasting by simultaneously disentangling multi-scale temporal patterns and learning adaptive spatial structures before sequence modeling. Unlike previous approaches that apply LLMs to raw, entangled time series or rely on static graphs, we leverage Variational Mode Decomposition (VMD) as a pre-processing component to separate the time series into intrinsic modes before the LLM models temporal patterns. It allows the model to capture intrinsic features like trend, daily patterns, and high-frequency fluctuations. Furthermore, we propose a dynamic spatial learning method that is mode-dependent and learns the graph topology of different temporal modes individually, since the spatial connection varies across the frequencies. For instance, the correlation is not similar between the long-term trend and high-frequency behaviors. Our model can learn the decomposed temporal modes with their respective spatial connections, which makes it more adept at capturing multi-scale information compared to other models using either raw data or static graphs.

The major contributions of this work are outlined as follows:

- In this paper, we propose a decomposition-based spatiotemporal forecasting framework that models multi-scale temporal patterns and their corresponding spatial dependencies, thereby addressing the limitations of existing methods that operate on entangled time series.

- Within this framework, we design a mode-dependent dynamic graph learning mechanism that constructs a separate graph for each decomposed temporal mode to capture the homogeneous patterns, enabling adaptive modeling of traffic interactions across different frequency components.

- We further integrate these mode-specific graphs into a graph-aware pretrained large language model, enabling it to leverage dynamic spatial structures while effectively capturing long-range temporal dependencies.

- We validate the effectiveness of the proposed framework through a comprehensive experimental evaluation across various benchmark real-world datasets, showing significant improvements in both short- and long-term forecasting scenarios.

The proposed DG-LLM model is evaluated on six real-world datasets spanning both grid and graph-structured traffic networks in short and long forecasting tasks. Our framework shows strong superiority compared to existing approaches with average improvements in terms of MAE and RMSE of $13\% - 19\%$ and $19\% - 25\%$, respectively. Besides, we perform several experiments to investigate the robustness of our model with respect to ablations, missing data, zero-shot generalization from one dataset to another, and efficiency. Notably, our model performs remarkably well on long-term forecasting scenarios where error propagation occurs for earlier methods due to their inability to capture evolving patterns over time. This is because our model leverages temporal dependencies along with evolving spatial interactions between nodes within the network.

The remainder of this paper is organized as follows. The Related Work section discusses previous works on graph-based, LLM-based, and decomposition-based approaches. The Problem Statement section introduces problem formulation and notations. The Methodology section presents the proposed DG-LLM framework. In the Experiments section, we discuss our dataset, baselines, evaluation metrics, and experimental setup. The Results section presents the experimental results for both short-term and long-term forecasting, along with ablation studies and additional analyses. The Discussion section presents the findings, and the Conclusion section concludes the paper, outlining future research directions.

## Related work

Traffic forecasting has been extensively studied from different perspectives due to its importance in ITS. Early research focused on classical statistical models such as ARIMA [5] and Kalman Filters [6], whose study of temporal dependence opened up the way for later methods. As urban data became more complex, subsequent studies shifted toward architectures that were able to capture the non-linear and high-dimensional spatiotemporal dependencies. This section reviews

prior research on traffic forecasting from three major perspectives: graph-based, LLM-based, and decomposition-based approaches.

## 2.1. Graph-based architectures

The graph-based deep learning architecture has become the dominant paradigm used in traffic prediction due to the inclusion of road network information and the application of Graph Neural Networks (GNNs) to represent spatial inter-dependencies among different road segments. Early traffic prediction studies have employed traditional deep learning approaches like autoregressive neural networks, CNNs, and RNNs, where traffic information is treated as a sequence of independent time series or grid representation. These methods are effective in handling temporal information but lack the capacity to represent the complex topological structure of the road network information. The evolution of the Graph Convolutional Networks (GCNs) allowed the modeling of the non-Euclidean spatial relationship, thus becoming an important breakthrough in urban computing. Spatiotemporal GNNs have been able to overcome previous challenges through the use of graph convolutional layers in deep learning architectures, which can be broadly categorized by their temporal modeling strategies: CNN-based, RNN-based, and attention-based frameworks.

CNN-based architectures, including STGCN [9], Graph WaveNet (GWN) [12], and MTGNN [13], incorporate 1-D temporal convolutional layers along with graph convolutional layers. These architectures are computationally efficient and maintain gradient stability by stacking dilated temporal convolutional layers. However, these architectures are limited to local windows for analysis as the convolutional kernel size is fixed, which restricts their ability to capture overall network patterns. Additionally, these models cannot accommodate dynamically varying traffic flow or changing road networks, as the convolutional layers inherently assume that the network is static. On the other hand, RNN-based architectures, including T-GCN [3], DCRNN [8], and AGCRN [14], extend recurrent units such as GRUs or LSTMs by replacing fully connected operations with graph convolutions. As a result, these models are capable of capturing localized spatial dependencies from neighboring sensors at each recurrence step. Nevertheless, this process is computationally expensive and prone to gradient vanishing issues, which makes it difficult to learn long-term dependencies effectively.

In order to solve these issues, attention-based architectures like GMAN [15] and ASTGCN [16] employ attention layers, which assign different weights to various time steps and spatial neighbors. It can efficiently learn global context information. Yet, it comes with the cost of high parameter complexity, thus necessitating high computational and memory requirements. In addition, Recent advances have attempted to address the predefined graph structure issue. For example, A3TGCN [17] extends T-GCN [3] by applying the attention module to discover global traffic flow tendencies. Furthermore, recent studies have explored robustness and continuous spatiotemporal modeling to move past the static graph assumptions. PSTGCN [2] employs a dynamic probabilistic spatiotemporal graph model to address the static adjacency problem, whereas R-PST-GCN [18] expands on this framework to enhance its robustness to noisy data. DGCRN [19] and DMST-GCN [20] incorporate hyper-networks and adaptive modules to dynamically learn evolving adjacency matrices. In addition, other methods such as curriculum learning have also been applied to optimize the stability of the optimization procedure to reduce error accumulation during training [19]. Other novel approaches, like STG-NCDE [21], have employed neural differential equations to overcome the challenges faced due to discrete sampling in order to improve the representation of temporal continuity.

Despite these advances, most graph-based methods rely on predefined or weakly adaptive adjacency structures, limiting their ability to capture frequency-dependent and evolving spatial interactions. In addition, these methods typically require training on each dataset separately, which hinders their capacity to learn temporal patterns in small datasets.

## LLM-Based architectures

Recent studies have shown that transformer architectures [22] can achieve strong performance in time-series forecasting by effectively learning long-range dependencies. Traditional transformer-based models, such as Informer [23] and

Autoformer [24], have improved the scalability of time-series modeling through sparse attention mechanisms. Nonetheless, these models usually need extensive datasets when trained from scratch, making them limited in scalability and adaptability. Hence, recent studies have shifted towards adapting pretrained Large Language Models (LLMs) for forecasting by converting numerical sequences into embedding representations [7]. This approach allows utilizing enormous knowledge of the pretrained LLMs for temporal modeling in time series forecasting. Frameworks such as LLMTime [25] and Time-LLM [26] leverage the temporal modeling capabilities of models like GPT-2, demonstrating strong zero-shot and few-shot generalization. Other approaches, such as PromptCast [27], further explore prompt-based formulations by converting numerical time series into textual prompts for direct inference using LLMs, while models like LLM4TS [28] adopt fine-tuning strategies to adapt pretrained language models for time-series forecasting. However, these models primarily focus on sequential patterns and overlook spatial dependencies.

There are emerging LLM architectures like UrbanGPT [29] and UniTime [30] that integrate spatial context using instruction tuning and in-context learning. Temporal models, including TFT [31] and UrbanMind [32], achieve high efficiency and effectiveness in predicting multiple time horizons. Even though these methods exhibit great improvements, these approaches cannot be easily extended to dynamic spatial graphs. Spatiotemporal LLMs, like STLLM [7], partially address the transferability and efficiency of LLMs by finetuning a subset of the model's parameters while incorporating node representations. STLLM+ [10] extends this design by integrating graph-based spatial information, while GATLLM [33] further improves performance by incorporating graph attention mechanisms to jointly model spatial dependencies within the LLM framework. Furthermore, contemporary approaches such as Vision-LLMs [34] explore vision-language fusion to capture global spatial representations from image data, suggesting that transforming spatiotemporal data into visual modalities can enhance long-horizon forecasting.

Although LLM-based approaches exhibit robust capabilities in terms of modeling long-term temporal dependencies and generalization, most efforts focus on modeling sequences without considering spatial dependencies in a dynamic way or their multi-scale nature. This necessitates a shift toward methods that can effectively disentangle these complex signals to capture complex traffic dynamics across different frequency components.

## Decomposition-based architectures

One of the biggest challenges in traffic forecasting is dealing with the non-stationarity and multi-scale behavior of sensor data. This can be achieved through the use of signal decomposition, which has been widely employed as a technique for separating the complex and noisy signal into simpler and more predictable components in various domains. Signal decomposition has been used commonly to decompose time-series data into interpretable frequency components. Although wavelet transform-based methods [35] and Empirical Mode Decomposition (EMD) variants [36,37] offer data-driven flexibility, they are often hindered by mode-mixing and sensitivity to noise.

Variational Mode Decomposition (VMD) [11] has emerged as a robust alternative by formulating decomposition as an optimization problem, ensuring stable and well-separated frequency components. Recent studies have integrated VMD with deep learning architectures for traffic forecasting, such as LSTMs and encoder–decoder frameworks [38,39], to enhance feature extraction. However, these approaches mainly concentrate on the temporal decoupling but ignore how decomposed signals evolve in spatially interconnected networks. Although these techniques can help to enhance the modeling process by extracting temporal features, they are generally combined with traditional deep-learning models and do not explore how to utilize them with pretrained LLMs to capture multi-scale temporal patterns and long-range dependencies.

Overall, existing approaches address different aspects of traffic forecasting but remain fragmented. Graph-based models capture spatial dependencies without modeling multi-scale temporal patterns, LLM-based methods learn long-range temporal dependencies but lack dynamic spatial reasoning, and decomposition-based approaches improve temporal representation without incorporating spatial dynamics or pretrained sequence models. To the best of our knowledge, prior

studies have not jointly combined multi-scale temporal decomposition, dynamic spatial dependency modeling, and pre-trained sequence representations within a unified framework. Therefore, such an integration remains an open challenge.

## Problem statement

### Traffic network representation

We represent the traffic network as a weighted directed graph $\mathcal{G} = (\mathcal{V}, \mathcal{E}, \mathbf{A})$, with $\mathcal{V}$ being a set of $N = |\mathcal{V}|$ nodes that stand for sensors or road segments, and $\mathcal{E}$ being the set of edges that show how they are physically or functionally connected. The adjacency matrix, $\mathbf{A} \in \mathbb{R}^{N \times N}$, shows how nodes are related to each other. A feature matrix $\mathbf{X}_t \in \mathbb{R}^{N \times F}$ shows the state of the network at any given time step $t$. It combines $F$ traffic metrics (like speed, flow, or demand) from all $N$ nodes.

Given an input window of $T_{\text{in}}$ time steps, the model observes a spatiotemporal sequence and predicts future traffic conditions over the next $T_{\text{out}}$ steps. This forecasting task can be written as:

$$\hat{Y}_{t+1:t+T_{\text{out}}} = f_\theta \left( X_{t-T_{\text{in}}+1:t} \right),$$

(1)

where $f_\theta$ denotes the learned prediction function, $X_{t-T_{\text{in}}+1:t}$ represents the input traffic sequence, and $\hat{Y}_{t+1:t+T_{\text{out}}}$ represents the predicted traffic states.

### Notations

A summary of frequently used symbols and their definitions is provided in Table 1. The concatenation operator is written as $[\cdot \| \cdot]$, and $(\cdot)^\top$ denotes the transpose operation. The background of the key concepts related to VMD, graph neural networks, and Transformer architecture with LoRA is included in S1 Appendix.

## Methodology

In this section, we present our proposed framework named DG-LLM: **D**ecomposition-Based Dynamic **G**raph Adaptation of **L**arge **L**anguage **M**odels for Spatiotemporal Traffic Forecasting, as shown in Fig 2. We begin with an overview of the proposed framework, followed by a description of its key components.

**Table 1. Summary of Notation Used in the Framework.**

| Symbol | Description |
|---|---|
| $A$ | Static adjacency matrix |
| $\mathcal{G}$ | Traffic network graph structure |
| $V$ | Set of Nodes |
| $N$ | Number of Nodes (sensors or road segments) |
| $\mathcal{E}$ | Set of Edges |
| $F$ | Number of Features per node |
| $T_{\text{in}}$ | Length of input sequence |
| $T_{\text{out}}$ | Forecasting horizon |
| $X$ | Input observation sequence |
| $Y$ | Future traffic sequences |
| $K$ | Number of VMD modes |
| $C$ | Embedding dimension for feature representations |
| $H$ | Hidden dimension of the LLM |
| $U_k$ | $k$-th VMD mode across all nodes |
| $E_k$ | Spatiotemporal embedding for mode $k$ |

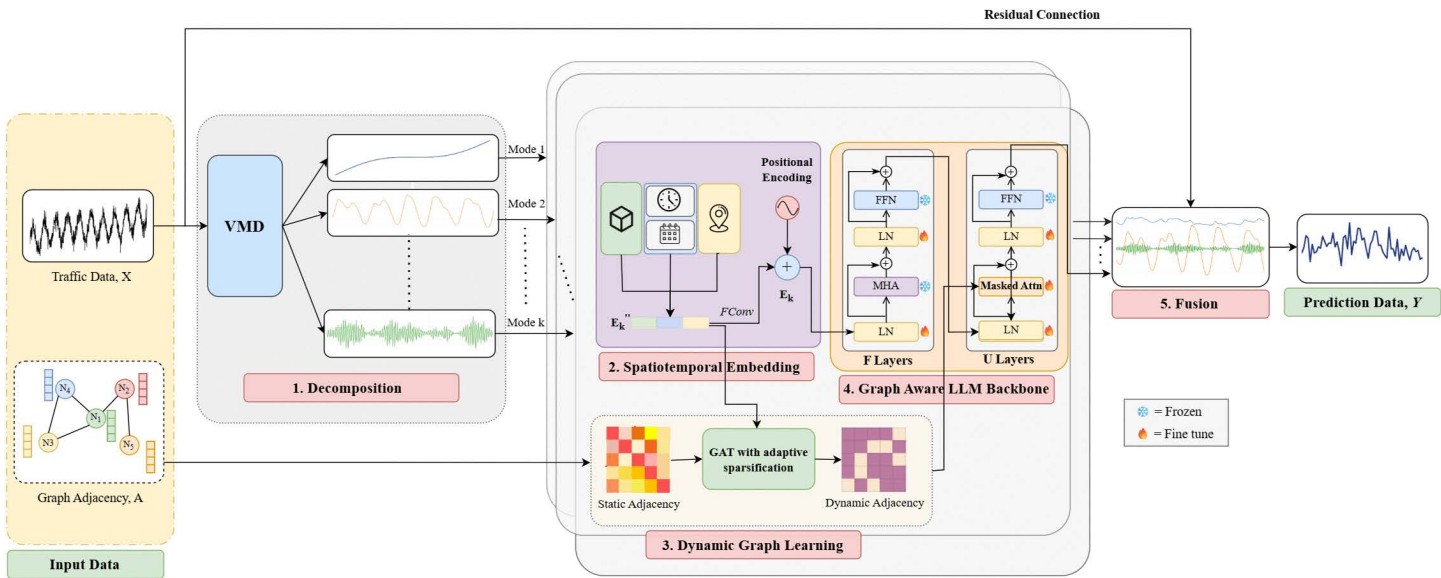

**Fig 2. Overall architecture of the proposed Decomposition-based Dynamic Graph Adaptation of Large Language Models for Spatiotemporal Traffic Forecasting (DG-LLM) framework.**

## Framework overview

The proposed DG-LLM framework is composed of five major components: (i) Variational Mode Decomposition, (ii) Spatiotemporal Embedding, (iii) Dynamic Graph Learning, (iv) Graph-aware pretrained LLM Backbone, and (v) Multi-Mode Fusion.

The workflow starts with **Variational Mode Decomposition (VMD)**, which decomposes the traffic signal data into multiple modes. These modes include specific patterns such as diurnal variations, weekly trends, and traffic spikes. Hence, the model will be able to analyze specific patterns without any hindrances from other attributes. Each mode is then transformed into a **spatiotemporal embedding component** that combines traffic data, each node's spatial identity, and time-based context (e.g., hour of day, day of week). This embedding ensures that the model considers both temporal and spatial aspects of the data, leading to more accurate predictions. After the embedding process, a **dynamic graph** will be created to learn the relationships between each node based on the modes. This will then be converted into a mask, guiding the LLM's attention mechanism to focus only on the highly connected nodes. It helps the model better understand how different locations in the traffic network interact. The spatiotemporal embedding and dynamic graph are then fed into the **pretrained LLM backbone**, where the data is further processed for prediction. After processing all modes, the model combines the LLM results with a residual projection of the original input using a **fusion mechanism**. Through the integration of these five components, the model can accurately capture the complex temporal patterns of traffic, respond to changing spatial relationships, and provide precise traffic forecasts.

## Variational mode decomposition

The heterogeneous temporal behavior of traffic flow exhibits both long- and short-term characteristics. To identify these characteristics, we propose using Variational Mode Decomposition (VMD) [11]. This method can decompose each time series at each node into intrinsic K modes, each corresponding to a specific frequency band. In general, low-frequency modes are associated with long-term characteristics, while intermediate- and high-frequency modes are associated with periodic and short-term characteristics, respectively.

Given an input window $X \in \mathbb{R}^{T_{in} \times N \times F}$, VMD is applied independently to each node's time series data (for each feature):

$$\mathbf{x}_n = \sum_{k=1}^{K} \mathbf{u}_{n,k},$$

$$\text{(2)}$$

where $\mathbf{x}_n \in \mathbb{R}^T$ represents the time series at node $n$, and $\mathbf{u}_{n,k} \in \mathbb{R}^T$ is the $k$-th intrinsic mode. This produces a set of mode-specific tensors:

$$U_k \in \mathbb{R}^{T_{in} \times N \times F}, \qquad k = 1, \ldots, K.$$

Each mode represents the dynamics of traffic on a particular time scale. This way, the model can process traffic patterns individually. For instance, the low-frequency modes capture the large-scale mobility of traffic, whereas the high-frequency modes capture abrupt changes in traffic caused by incidents or signal changes.

The individual modes are then taken as separate inputs to the spatiotemporal embedding and the backbone of the large language model (LLM). This way, the pre-trained LLM can mitigate interference between traffic dynamics at different frequencies.

### Spatiotemporal input embedding

The spatiotemporal input embedding layer transforms the decomposed traffic modes into a unified representation that incorporates three contexts: mode-specific traffic features, node information, and temporal periodicity.

Instead of using conventional text-based tokenization methods and word embeddings, feature vectors are directly constructed as input embedding to use in the pretrained model. The token embedding encodes the observed traffic features at each node, the spatial embedding identifies which node the data corresponds to, and the temporal embedding represents when the observation occurs through day-of-week and hour-of-day information.

**Token Embedding:** A point-wise convolution is performed on each intrinsic mode of traffic, $U_k \in \mathbb{R}^{T \times N \times F}$, to project the traffic features into a hidden dimension $C$:

$$E_k^{\text{token}} = \text{PConv}(U_k; \theta_p)$$

$$\text{(3)}$$

where $\theta_p$ denotes learnable parameters. This embedding captures the mode-specific traffic state at each node.

**Spatial Embedding:** To provide the model with each node's identity within the network, we design a learnable embedding shared across all modes:

$$E^{\text{spatial}} = \sigma(W_S \cdot U_k + b_s)$$

$$\text{(4)}$$

where $W_S$ and $b_s$ are trainable parameters and $\sigma(\cdot)$ is an activation function. This embedding provides a stable spatial reference independent of traffic variations.

**Temporal Encoding:** Traffic patterns vary periodically across time. To capture this, hour-of-day $X_{\text{day}} \in \mathbb{R}^{N \times T_d}$ and day-of-week $X_{\text{week}} \in \mathbb{R}^{N \times T_w}$ indicators are encoded using learnable linear projections:

$$E^{\text{time}} = W_d(X_{\text{day}}) + W_w(X_{\text{week}})$$

$$\text{(5)}$$

Here, $W_d$ and $W_w$ are learnable parameters. This helps the model differentiate between different traffic conditions at different times or on different days.

**Embedding Fusion:** To create a spatiotemporal embedding representation, the token embedding, spatial embedding, and temporal embedding are concatenated along the feature dimension:

$$E'_k = [E_k^{\text{token}} \| E^{\text{spatial}} \| E^{\text{time}}] \in \mathbb{R}^{N \times 3C} \tag{6}$$

A fusion convolution projects the combined representation into the LLM hidden dimension $H$:

$$\tilde{E}_k = \text{FConv}(E'_k; \theta_f) \in \mathbb{R}^{N \times H} \tag{7}$$

where $\theta_f$ represents the learnable parameters of the fusion convolution. Finally, a non-linear activation function (Leaky ReLU) is applied to the output of the fusion convolution $\tilde{E}_k$ to incorporate non-linearities in the model:

$$E''_k = \text{LeakyReLU}(\tilde{E}_k) \tag{8}$$

**Positional Encoding:** To maintain the temporal order of the input sequence, a learnable positional encoding $P_t$ is incorporated. This allows the model to understand the sequential order of the input and maintains compatibility with the pretrained model architecture of the LLM, resulting in the final input:

$$\mathbf{E}_k = E''_k + P_t \tag{9}$$

The resulting embedding $\mathbf{E}_k$ serves as the spatiotemporal input to the subsequent pretrained LLM backbone for traffic prediction.

## Dynamic graph learning

The major benefit of the proposed framework is its ability to directly learn dynamic, mode-specific connectivity from the data, rather than relying on a static adjacency matrix. The interaction between the traffic components is dynamic due to the varying traffic conditions, making the predefined road graph an unrealistic approximation. However, to address the aforementioned problem of the predefined road graph, a novel dynamic graph learning module is proposed as shown in Fig 3. The module (i) learns the spatial dependencies directly from the evolving temporal patterns, (ii) stabilizes training by gradually transitioning from a physical graph to a learned topology, and (iii) enforces sparsity to reduce the computational complexity while retaining the significant connections. The resulting graph is later used to constrain the LLM backbone's attention mechanism to ensure that spatial dependencies are modeled realistically and adaptively.

To model the complexities of real-world traffic, we employ the Graph Attention Network (GAT) mechanism [40] that learns how different locations (nodes) influence one another. By applying spatial self-attention, each node can selectively aggregate the information from the most relevant neighboring nodes.

We first project the node features into a higher-dimensional space using a shared weight matrix $\mathbf{W}$. To determine how much one location influences another, we calculate an attention coefficient $e_{ij}$ using a single-layer feedforward network with a LeakyReLU activation:

$$e_{ij} = \text{LeakyReLU}\left(\vec{\mathbf{a}}^\top [\mathbf{W}\mathbf{h}_i \| \mathbf{W}\mathbf{h}_j]\right) \tag{10}$$

To make these scores comparable across the network, we apply a softmax operation. This results in the normalized importance scores $\alpha_{ij}$ that define our dynamic adjacency matrix, $\mathbf{A}_{\text{learned}}$:

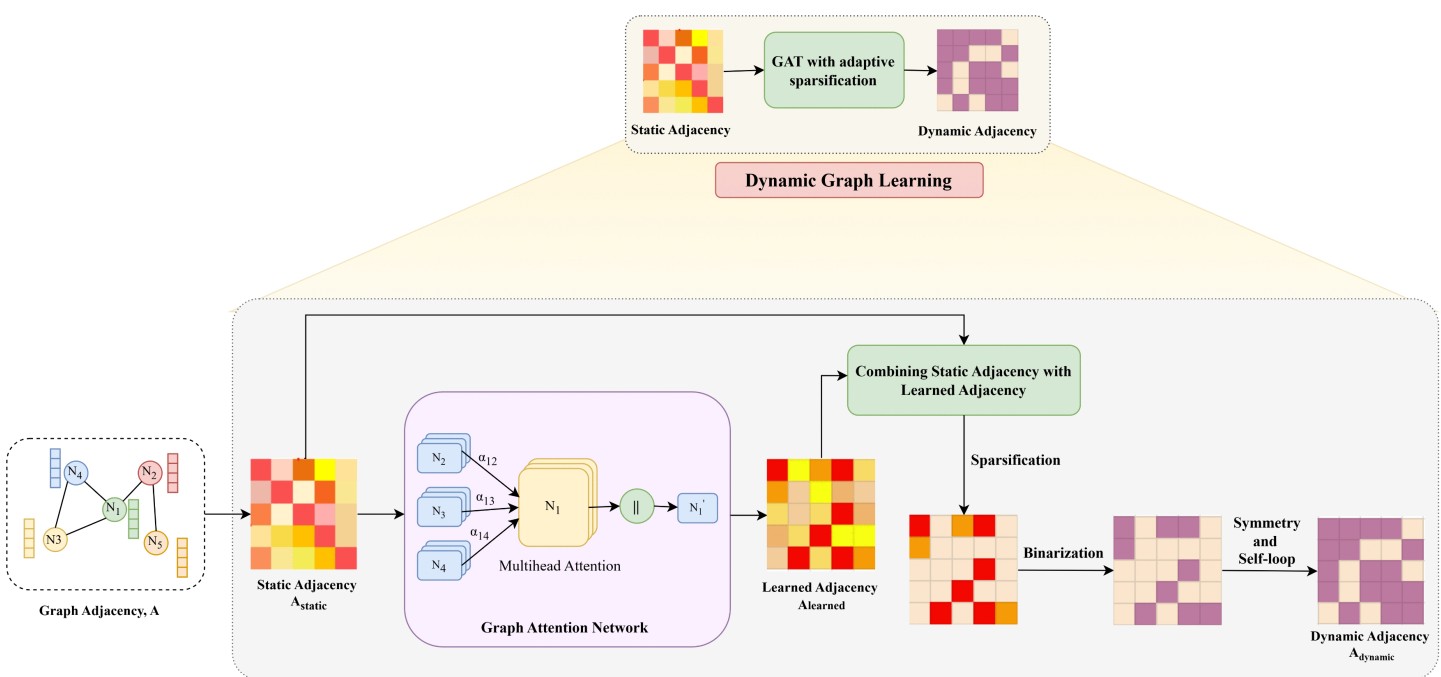

**Fig 3. Dynamic Graph Learning Pipeline for Mode-Specific Adjacency Construction.**

$$\mathbf{A}_{\text{learned}}(i,j) = \frac{\exp(e_{ij})}{\sum_{k=1}^{N} \exp(e_{ik})} \tag{11}$$

Since each intrinsic mode encodes distinct temporal characteristics, the learned adjacency adapts to the spatial dependencies for each temporal scale.

Learning $A_{\text{learned}}$ directly from evolving embeddings can be unstable during early training. To mitigate this, we adopt a curriculum-inspired training strategy that starts with the static adjacency and gradually introduces the learned patterns. The learned graph ($A_{\text{learned}}$) is integrated with the original static road network ($A_{\text{static}}$), by a blending coefficient, $\alpha$, that decreases over epochs:

$$A_{\text{blend}} = (1 - \alpha) A_{\text{learned}} + \alpha A_{\text{static}}, \tag{12}$$

The integration shifts the focus from physical topology to data-driven dependencies. Noise reduction and increased computational efficiency are achieved by sparsification, which applies a threshold $\tau$ for pruning weak connections:

$$A_{\text{sparse}}(i,j) = \begin{cases} A_{\text{blend}}(i,j), & A_{\text{blend}}(i,j) \geq \tau, \\ 0, & \text{otherwise.} \end{cases} \tag{13}$$

The remaining edges are binarized and symmetrized to represent the bidirectional nature of traffic flow. The identity matrix ($I_N$) is added to include self-loops in the network:

$$A_{\text{dynamic}} \leftarrow \max(A_{\text{dynamic}}, A_{\text{dynamic}}^{\top}) + I_N \tag{14}$$

The binary matrix, $\mathbf{A}_{\text{dynamic}}$, represents the final dynamic spatial topology. The matrix is converted to a graph mask and fed into the LLM's attention mechanism. The attention mechanism is constrained by the learned spatial dependencies. The algorithm for learning the dynamic graph is shown in Algorithm 1.

## Algorithm 1 Dynamic Graph Learning

```
Input: Static adjacency A_static, node features {h_1,...,h_N}, projection matrix W, blending coefficient
α, pruning threshold τ
Output: Final dynamic adjacency matrix A_dynamic
1: z_i ← Wh_i for all i ∈ {1,...,N}                          ▷ Linear Projection
2: for each node pair (i, j) do
3:     e_ij ← LeakyReLU(a⃗ᵀ[z_i ‖ z_j])                      ▷ Unnormalized attention score
4: end for
5: for each node i do
6:     A_learned(i, j) ← exp(e_ij) / ∑_{k=1}^N exp(e_ik)      ▷ Normalized scores via Softmax
7: end for
8: A_blend ← (1 − α) A_learned + α A_static                   ▷ Curriculum Learning
9: for each pair (i, j) do
10:    if A_blend(i, j) ≥ τ then
11:       A_sparse(i, j) ← A_blend(i, j)                      ▷ Sparsification (Pruning))
12:    else
13:       A_sparse(i, j) ← 0
14:    end if
15: end for
16: A_dynamic(i, j) ← 1 if A_sparse(i, j) > 0 else 0          ▷ Binarization
17: A_dynamic ← max(A_dynamic, A_dynamicᵀ) + I_N             ▷ Symmetrization with Self-Loop
```

## Graph-Aware Pretrained LLM Backbone

The spatiotemporal embedding of each intrinsic mode is then processed through a graph-aware pre-trained LLM backbone. While pre-trained LLMs are efficient at capturing long-range temporal dependencies via self-attention, they generally do not capture topological constraints common in road networks. This is achieved by integrating a learned dynamic graph into the model's attention mechanism, enabling it to focus on spatially relevant interactions rather than arbitrary global contexts. As shown in Fig 4, each intrinsic mode is associated with its distinct learned dynamic graph that dictates its unique attention mechanism within the LLM backbone.

We utilize GPT-2 as the primary backbone model due to its robust autoregressive prediction performance. To balance the retention of existing temporal information with adapting to traffic-specific information, the $L$ layers of the transformer are split into two distinct groups: The first $F$ layers are frozen to ensure that the primary temporal representations learned during pretraining are preserved. In these layers, information flow is based on standard causal self-attention [22]:

$$\bar{H}^i = \text{MHA}(\text{LN}(H^{i-1})) + H^{i-1}, \tag{15}$$

$$H^i = \text{FFN}(\text{LN}(\bar{H}^i)) + \bar{H}^i, \tag{16}$$

where $H^0$ denotes the mode-specific spatiotemporal embedding augmented with positional encoding.

Spatial dependencies are explicitly introduced in the upper $U$ trainable layers. In this stage, the learned dynamic adjacency matrix $\mathbf{A}_{\text{dynamic}}$ is utilized to construct a graph mask, $M_{\text{graph}}$, which is then fused with the standard causal mask $M_{\text{causal}}$:

$$M_{\text{combined}} = M_{\text{graph}} + M_{\text{causal}} \tag{17}$$

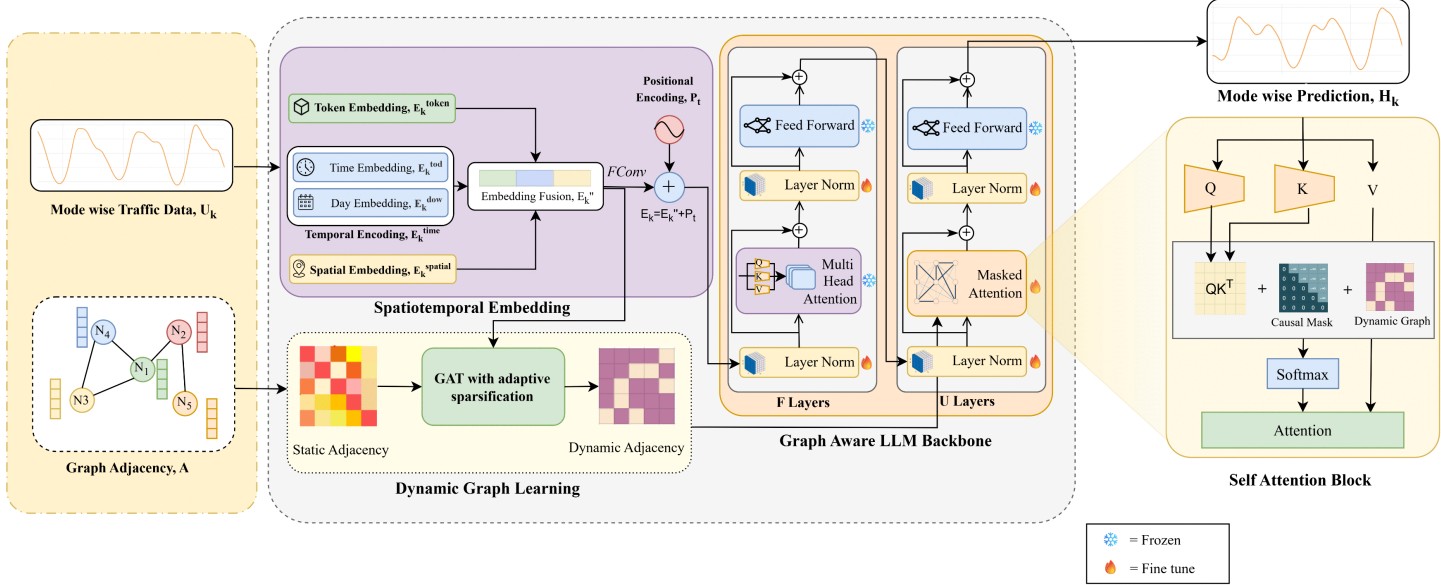

**Fig 4. Mode-wise data processing through spatiotemporal embedding with graph-aware LLM backbone.**

The attention operation in these layers becomes:

$$\bar{H}^i = \text{MHA}(\text{LN}(H^{i-1}), M_{\text{combined}}) + H^{i-1}, \tag{18}$$

which restricts information flow to spatially connected nodes while preserving autoregressive ordering.

To adapt the backbone to the traffic dynamics without the computational cost of fine-tuning the model, the Low-Rank Adaptation (LoRA) technique [41] is used to update the query ($W_Q$) and key ($W_K$) projections of the unfrozen layers. The weights are updated via low-rank matrices:

$$\triangle W_Q = B_Q A_Q, \qquad \triangle W_K = B_K A_K, \tag{19}$$

where $W_Q, W_K \in \mathbb{R}^{H \times H}$, $A_Q, A_K \in \mathbb{R}^{r \times H}$ and the rank $r \ll H$. The adapted projections are:

$$W'_Q = W_Q + \triangle W_Q, \qquad W'_K = W_K + \triangle W_K. \tag{20}$$

This reduces the number of trainable parameters while preserving the model's ability to learn traffic-specific attention patterns. The final attention computation for a given mode $k$ is defined as:

$$\text{Attention}(E_k) = \text{softmax}\left( \frac{(E_k W'_Q)(E_k W'_K)^\top}{\sqrt{d_k}} + M_{\text{combined}} \right)(E_k W_V), \tag{21}$$

This allows the model to learn the attention patterns for each mode while keeping the value projections ($W_V$) frozen, thereby ensuring stability. The resulting output representation $H_k^{\text{out}}$ is forwarded to the multi-mode fusion stage for final prediction.

## Multi-mode fusion

The output of the graph-aware LLM is a set of mode-specific representations, given by:

$$\{H_k^{\text{out}}\}_{k=1}^K, \qquad H_k^{\text{out}} \in \mathbb{R}^{N \times H}.$$

To capture the distinct characteristics of each VMD mode, we first project these representations onto the prediction horizon using independent 1D convolutional layers:

$$\hat{Y}_k = \text{Conv1D}\left(H_k^{\text{out}}; W_k^{\text{proj}}\right) \in \mathbb{R}^{T_{\text{out}} \times N}, \tag{22}$$

applied for each mode $k \in \{1, \dots, K\}$. These representations need to be fused together, as they are derived from the original signal that was decomposed into various modes using VMD. Therefore, these individual predictions are then fused into a unified base representation using learned mode-specific weights:

$$\hat{Y}_{\text{base}} = \sum_{k=1}^K w_k \, \hat{Y}_k, \tag{23}$$

where $w_k$ are scalar weights normalized via a softmax function to ensure $\sum_{k=1}^K w_k = 1$:

$$w_k = \frac{\exp(\alpha_k)}{\sum_{j=1}^K \exp(\alpha_j)}, \tag{24}$$

with $\{\alpha_k\}_{k=1}^K$ being learnable parameters. This representation combines information over various scales, capturing both long- and short-term information of the traffic signal.

A residual connection is added to this process, where the final time step of the input is directly projected onto the output horizon. The final forecast is defined as:

$$\hat{Y} = \hat{Y}_{\text{base}} + \lambda \, \hat{Y}_{\text{residual}}, \tag{25}$$

where $\lambda$ is a learned scalar parameter. The residual connection improves the stability of our forecasts by directing predictions towards the most recent observations.

The complete algorithm is shown in 2.

## Algorithm 2 DG-LLM Framework

```
Input: Traffic sensor data X, Static adjacency matrix A
Output: Forecast Ŷ
1: for each node n in parallel do
2:    {U₁,...,U_K} ← VMD(X_:,n)                          ▷ Multi-Scale Decomposition
3: end for
4: for each mode k ∈ {1,...,K} do
5:    Token, Spatial and Temporal embedding using Eq. (3), (4), and (5)
6:    E'_k ← [E_k^token ‖ E^spatial ‖ E^time]            ▷ Embedding Fusion using Eq. (6)
7:    E''_k ← LeakyReLU(FConv(E'_k))
8:    E_k ← E''_k + P_t                                  ▷ Positional Encoding using Eq. (9)
9:    A_k^dynamic ← DynamicGraphLearning(E_k, A)         ▷ See Alg. 1
10:   H⁰ ← E_k
11:   for l=1 to L do
```

```
12:     H^l ← LayerNorm(H^{l-1})
13:     if l ≤ F then                           ▷ Frozen Pre-trained Layers
14:        H^l ← MHA(H^l) + H^{l-1}
15:     else                                    ▷ Graph-Injected Adaptive Layers
16:        Construct graph mask: M_k ← Mask_causal ⊙ A_k^dynamic
17:        H^l ← MaskedAttn(H^l, M_k) + H^{l-1}
18:     end if
19:     H^l ← FFN(LayerNorm(H^l)) + H^l
20:   end for
21:   H_k^out ← H^L
22: end for
23: Ŷ_base ← Σ_{k=1}^{K} w_k Ŷ_k                 ▷ Fusion (Eq. 23)
24: Ŷ ← Ŷ_base + λŶ_res                          ▷ Final Weighted Forecast (Eq. 25)
25: return Ŷ
```

## Experiments

### Datasets

We evaluate the proposed framework on six real-world traffic datasets, including both grid-structured urban mobility systems and graph-structured highway traffic networks. This variety allows for a comprehensive assessment across different spatial representations, traffic modalities, and temporal resolutions. Statistics on the datasets are provided in Table 2.

**Grid-Based Urban Mobility** NYC-Taxi [42] and CH-Bike [43] are large-scale urban mobility datasets collected from the New York City taxi records and Citi Bike trip history. These datasets are aggregated over spatial grids and are therefore in a structured format in which demand is represented at a particular coordinate on the map. Both datasets capture pickup and drop-off demand. These demands have a strong weekly and daily pattern.

**Graph-Based Highway Traffic** PEMS04 and PEMS08 are benchmark highway traffic datasets collected from the Caltrans Performance Measurement System (PeMS) in California [44]. Traffic sensors are mapped to irregular road network graphs, capturing non-Euclidean spatial dependencies. Each dataset provides multivariate traffic measurements, including flow, speed, and occupancy, recorded across large-scale sensor networks.

### Baselines

We compare our proposed method against a diverse set of strong baselines that represent the main research directions in traffic forecasting. These baselines include both graph-based spatiotemporal models and LLM-based forecasting models.

- **STGCN** [9]: A spatial-temporal graph convolutional network that merges graph convolutions with gated temporal convolutions to jointly capture spatial and temporal correlations.

- **GWN** [12]: A Graph WaveNet model that integrates adaptive graph convolutions with dilated causal temporal convolutions to model complex spatiotemporal dependencies.

**Table 2. Statistics of the traffic forecasting datasets used in this study.**

| Dataset | # Units | # Timestamps | Granularity | Time Span | Structure | Data Type |
|---------|---------|--------------|-------------|-----------|-----------|-----------|
| NYC-Taxi | 266 | 4,368 | 30 mins | Apr–Jun 2016 | Grid | D |
| CH-Bike | 250 | 4,368 | 30 mins | Apr–Jun 2016 | Grid | D |
| PEMS04 | 307 | 16,992 | 5 mins | Jan–Feb 2018 | Graph | F, S, O |
| PEMS08 | 170 | 17,856 | 5 mins | Jul–Aug 2016 | Graph | F, S, O |

D = Demand, F = Traffic Flow, S = Speed, O = Occupancy.

- **AGCRN** [14]: An adaptive graph convolutional recurrent network that employs node-specific parameters and adaptive graph learning to capture fine-grained spatiotemporal correlations.

- **TGCN** [3]: A temporal graph convolutional model that combines graph convolutional networks with GRUs to simultaneously capture spatial topology and temporal dynamics.

- **GMAN** [15]: A multi-attention-based encoder–decoder framework that utilizes spatiotemporal attention blocks to model the dynamic correlations essential for traffic forecasting.

- **ASTGCN** [16]: An attention-based spatiotemporal graph convolutional network that incorporates spatial–temporal attention with graph convolutions to capture dynamic traffic patterns.

- **STLLM** [7]: A spatial–temporal LLM that represents multi-location time series as token sequences and uses partial fine-tuning to adapt a pretrained LLM for forecasting tasks.

- **STLLM+** [10]: An extension of STLLM that integrates graph-based spatial information through attention mechanisms to enable spatially informed temporal modeling.

## Implementation details

All datasets, including NYC-Taxi, CH-Bike, and PeMS, are divided chronologically into training, validation, and test sets with a standard division of 6:2:2. The input sequence length and prediction horizon are set at $T_{in} = T_{out}$ = 12 and 96 time steps, respectively, to test the model's performance in short- and long-term forecasting. Temporal encoding is based on periodicity in each dataset. For the NYC-Taxi and CH-Bike datasets, the daily periodicity $T_d$ is set to 48 time steps, with a sampling time of 30 minutes. For the PeMS dataset, the daily periodicity, $T_d$, is set to 288 time steps with a sampling time of 5 minutes. For all datasets, the weekly periodicity, $T_w$, is set to 7.

To improve computational efficiency, VMD is performed with $K = 3$ modes, a bandwidth constraint of $\alpha$ = 2000, and a convergence tolerance of $1 \times 10^{-7}$. VMD is applied to each window separately using only the input sequence $X$ of length $T_{in}$, without including any future prediction window $Y$ of length $T_{out}$ to avoid any future information leakage. The core architecture utilizes a six-layer GPT-2 backbone, fine-tuned via LoRA with a rank ($r$) of 16 and an alpha of 32. To maintain spatial efficiency in dynamic graph learning, we apply a sparsity threshold $\tau$ that retains the top 15% of learned edges per node. Activation checkpointing is further employed to reduce GPU memory consumption by recomputing intermediate activations during backpropagation.

All experiments on the proposed model and baselines are conducted on NVIDIA Tesla T4 GPUs (16GB memory) using mixed-precision training. The model is optimized using the Ranger optimizer with a learning rate of 0.001. Due to GPU memory constraints, the batch size is set to 8, and all models are trained for 100 epochs. At the start of each epoch, the training data is randomly shuffled to improve generalization.

## Evaluation metrics

We use three standard metrics to evaluate the accuracy of our models: the Mean Absolute Error (MAE), the Mean Absolute Percentage Error (MAPE), and the Root Mean Squared Error (RMSE). For the ground truth value $Y_i$ and our prediction value $\hat{Y}_i$, given that we have $m$ data points,

$$\text{MAE} = \frac{1}{m} \sum_{i=1}^{m} |\hat{Y}_i - Y_i|$$

(26)

$$\text{MAPE} = \frac{100}{m} \sum_{i=1}^{m} \left| \frac{\hat{Y}_i - Y_i}{Y_i} \right| \tag{27}$$

$$\text{RMSE} = \sqrt{\frac{1}{m} \sum_{i=1}^{m} (\hat{Y}_i - Y_i)^2} \tag{28}$$

Lower values represent better results. The criterion chosen for the model optimization is the MAE due to its robust nature and stability to outliers. It calculates the average difference between the predicted and actual values, regardless of the direction of the error.

## Results

In this section, experimental results and analyses for six different traffic datasets are provided in order to evaluate the effectiveness of the proposed framework concerning the research questions stated below:

- **RQ1**: How well does the proposed model perform in comparison to the state-of-the-art methods in short-term forecasting?

- **RQ2**: How effectively does the proposed model sustain forecasting accuracy in long-term forecasting relative to other state-of-the-art models?

- **RQ3**: What is the impact of different components of the model on its forecasting effectiveness?

- **RQ4**: How do key hyperparameters and backbone choices influence forecasting accuracy?

- **RQ5**: To what extent does increasing data sparsity degrade forecasting performance?

### RQ1. Short-term forecasting

The quantitative comparison results against baseline methods are reported in Tables 3–5. Table 3 and 4 present traffic forecasting performance across all baseline models (for 30 minutes, 3 hours, 6 hours, and average) on the NYC-Taxi and CH-Bike datasets, respectively. To further validate our models, we compare them on the graph-based PeMS dataset (for 15 minutes, 30 minutes, 1 hour, and average), shown in Table 5. For each metric, the best-performing results are highlighted in bold, and the second-best results are underlined. Our model is run five times with deterministic seeds for reproducibility, and the results are reported as Mean±Standard Deviation (Std).

In both datasets, NYC-Taxi and CH-Bike, our model, DG-LLM, achieves the best performance across most prediction horizons. In NYC-Taxi, our model DG-LLM shows the lowest MAE and RMSE values, on average, for drop-off and pick-up datasets. For drop-off prediction at a 30-minute horizon, our model shows competitive performance in accuracy, with an MAE of 4.813 and an RMSE of 7.906 for the Pick-up dataset, and an MAE of 4.672 and an RMSE of 7.692 for the Drop-off dataset. As the horizon increases, other models' errors increase, while our model maintains consistent performance. At a 6-hour horizon, our model shows an MAE of 5.426 and an RMSE of 9.124 for the Pick-up dataset, and an MAE of 5.260 and an RMSE of 8.591 for the Drop-off dataset, outperforming all other models, including STLLM+, which shows an MAE of 5.550 and an RMSE of 9.981. The average over the 12-hour horizon shows the same trend: our model, DG-LLM, has the lowest MAE and RMSE among all other models for NYC-Taxi prediction.

The benefits of our proposed method are more visible in the CH-Bike data, which has even sparser, more local demand patterns. Our model outperforms other models with the lowest predicted error at all horizons for both Pick-up and Drop-off tasks. For example, at the 30-min horizon, our proposed method, DG-LLM, has the lowest MAE and RMSE, 1.697 and

**Table 3. Short-Term comparison on NYC-Taxi Dataset (Pick-up and Drop-off).**

| Horizon | Models | NYC-Taxi Pick-up | | | NYC-Taxi Drop-off | | |
|---|---|---|---|---|---|---|---|
| | | MAE ↓ | RMSE ↓ | MAPE ↓ | MAE ↓ | RMSE ↓ | MAPE ↓ |
| Horizon 1 (30 mins) | STGCN | 5.104 | 8.942 | 36.62% | 4.871 | 9.690 | 37.10% |
| | TGCN | 6.813 | 11.303 | 56.79% | 6.701 | 11.296 | 58.14% |
| | ASTGCN | 4.950 | 8.600 | 35.50% | 4.890 | 8.580 | 38.61% |
| | ST-LLM | 4.999 | 8.308 | 38.37% | 4.754 | 7.748 | 37.17% |
| | STLLM+ | **4.722** | 7.939 | **31.78%** | **4.548** | **7.442** | **33.96%** |
| | Our Model | 4.813±0.071 | **7.906**±0.123 | 34.45±2.57% | 4.672±0.118 | 7.692±0.199 | 34.85±1.48% |
| Horizon 6 (3 hours) | STGCN | 5.715 | 10.162 | 37.68% | 5.478 | 10.645 | 39.52% |
| | TGCN | 9.138 | 17.009 | 70.54% | 9.005 | 19.820 | 62.36% |
| | ASTGCN | 5.710 | 10.380 | 38.63% | 5.850 | 11.070 | 42.29% |
| | ST-LLM | 5.627 | 10.103 | 36.94% | 5.532 | 9.958 | 36.06% |
| | STLLM+ | **5.250** | 9.352 | **34.02%** | 5.291 | 9.448 | 35.29% |
| | Our Model | 5.262±0.053 | **8.888**±0.131 | 34.30±0.83% | **5.069**±0.031 | **8.625**±0.103 | **35.06**±0.36% |
| Horizon 12 (6 hours) | STGCN | 6.296 | 11.686 | 42.06% | 6.270 | 12.623 | 45.07% |
| | TGCN | 11.135 | 20.560 | 86.24% | 11.170 | 23.707 | 79.91% |
| | ASTGCN | 6.650 | 12.490 | 43.69% | 6.950 | 14.480 | 47.66% |
| | ST-LLM | 5.840 | 10.340 | 43.12% | 5.720 | 10.777 | **35.67%** |
| | STLLM+ | 5.550 | 9.981 | **35.50%** | 5.416 | 9.874 | 36.57% |
| | Our Model | **5.426**±0.070 | **9.124**±0.112 | 35.97±0.98% | **5.260**±0.084 | **8.917**±0.136 | 36.93±0.73% |
| Average of 1-12 Horizons | STGCN | 5.732 | 10.268 | 38.68% | 5.523 | 10.851 | 39.83% |
| | TGCN | 9.167 | 16.768 | 71.35% | 9.030 | 18.965 | 65.56% |
| | ASTGCN | 5.770 | 10.590 | 39.14% | 5.880 | 11.240 | 42.66% |
| | ST-LLM | 5.594 | 9.826 | 41.18% | 5.363 | 9.680 | 35.50% |
| | STLLM+ | **5.223** | 9.228 | 34.98% | 5.125 | 9.081 | **34.99%** |
| | Our Model | **5.223**±0.035 | **8.774**±0.066 | **34.76**±1.00% | **5.060**±0.043 | **8.591**±0.075 | 35.70±0.33% |

2.532, respectively, for Pick-up demand prediction and 1.610 and 2.323, respectively, for Drop-off demand prediction. Similarly, our method has the lowest MAE and RMSE, 1.742 and 2.654, respectively, for Pick-up demand prediction and 1.631 and 2.400, respectively, for Drop-off demand prediction at the 6-hour horizon. The 12-horizon average further highlights this advantage, with DG-LLM achieving the lowest MAE and RMSE across both tasks.

On the other hand, traditional graph-based convolutional models, such as TGCN, have consistently reported the highest errors across both datasets, indicating the limitations of graph-based convolutional models in modeling complex urban mobility patterns. Although models that utilize the attention mechanism, such as ASTGCN, have shown improvements over traditional graph-based convolutional models, they still perform worse than models that integrate LLMs. Although recent models, such as ST-LLM and STLLM+, have reported improvements in forecast accuracy by leveraging LLM capabilities, our model has achieved even higher performance, with stable, accurate forecasts in both datasets: one for dense taxi demand and the other for sparse bike-sharing demand.

To further assess the effectiveness of our model on graph-structured traffic datasets, we compared DG-LLM with several spatio-temporal forecasting models on the PeMS dataset, as shown in Table 5. In the PeMS04 dataset, DG-LLM consistently achieves the best forecasting results across all other forecasting models for different prediction horizons. For example, when the average prediction results over all 12 different prediction horizons are considered, our model achieves an MAE of 18.809 and an RMSE of 28.271, which is better compared to the best forecasting results achieved by the baseline model STGCN, where the MAE and RMSE values are 19.498 and 30.923, respectively. A similar improvement

**Table 4. Short-Term comparison on CH-Bike Dataset (Pick-up and Drop-off).**

| Horizon | Models | CH-Bike Pick-up | | | CH-Bike Drop-off | | |
|---|---|---|---|---|---|---|---|
| | | MAE ↓ | RMSE ↓ | MAPE ↓ | MAE ↓ | RMSE ↓ | MAPE ↓ |
| Horizon 1 (30 mins) | STGCN | 2.036 | 3.370 | 56.36% | 1.922 | 2.956 | 51.82% |
| | TGCN | 2.279 | 3.842 | 63.20% | 2.168 | 3.393 | 60.81% |
| | ASTGCN | 1.790 | 3.110 | 60.91% | 1.640 | 2.690 | 59.38% |
| | ST-LLM | 2.024 | 3.193 | 50.39% | 1.884 | 2.789 | 50.86% |
| | STLLM+ | 1.939 | 2.971 | 51.83% | 1.821 | 2.655 | 51.13% |
| | Our Model | **1.697**±0.006 | **2.532**±0.018 | **50.06%**±1.11% | **1.610**±0.007 | **2.323**±0.022 | **47.07%**±1.98% |
| Horizon 6 (3 hours) | STGCN | 2.117 | 3.542 | 54.37% | 2.010 | 3.136 | 51.32% |
| | TGCN | 2.493 | 4.284 | 65.84% | 2.433 | 4.013 | 61.52% |
| | ASTGCN | 2.030 | 3.680 | 60.95% | 1.830 | 3.030 | 61.08% |
| | ST-LLM | 2.128 | 3.360 | 50.63% | 2.013 | 3.042 | 48.91% |
| | STLLM+ | 2.044 | 3.210 | 51.76% | 1.895 | 2.822 | 50.38% |
| | Our Model | **1.750**±0.015 | **2.731**±0.039 | **48.16%**±1.06% | **1.651**±0.011 | **2.417**±0.024 | **45.83%**±0.70% |
| Horizon 12 (6 hours) | STGCN | 2.288 | 3.810 | 56.66% | 2.188 | 3.449 | 53.68% |
| | TGCN | 2.584 | 4.442 | 67.21% | 2.523 | 4.183 | 61.89% |
| | ASTGCN | 2.130 | 3.810 | 64.38% | 1.970 | 3.330 | 63.52% |
| | ST-LLM | 2.171 | 3.439 | 51.22% | 2.000 | 3.020 | 50.01% |
| | STLLM+ | 2.065 | 3.228 | **51.12%** | 1.932 | 2.895 | 50.16% |
| | Our Model | **1.742**±0.008 | **2.654**±0.015 | **46.84%**±0.51% | **1.631**±0.013 | **2.400**±0.027 | **43.94%**±0.68% |
| Average of 1-12 Horizons | STGCN | 2.144 | 3.570 | 55.42% | 2.034 | 3.182 | 51.93% |
| | TGCN | 2.479 | 4.252 | 65.77% | 2.403 | 3.941 | 61.35% |
| | ASTGCN | 2.010 | 3.620 | 61.91% | 1.830 | 3.050 | 61.27% |
| | ST-LLM | 2.121 | 3.352 | 50.61% | 1.994 | 3.016 | 49.09% |
| | STLLM+ | 2.033 | 3.180 | 51.51% | 1.889 | 2.808 | 50.16% |
| | Our Model | **1.739**±0.011 | **2.649**±0.027 | **48.17%**±0.80% | **1.640**±0.009 | **2.399**±0.021 | **45.49%**±0.63% |

in results is observed when each prediction horizon is considered individually. On the PeMS08 dataset, our model also achieves the best results among all baseline models. It achieves the minimum average error across all 12 prediction horizons, with MAE and RMSE of 14.507 and 22.645, respectively. These results highlight the stability and robustness of DG-LLM when modeling large-scale traffic sensor networks.

From the baseline comparisons, it is evident that convolutional graph models such as STGCN and Graph WaveNet perform better than RNN-based models such as TGCN and AGCRN. This shows that convolutional operations are more powerful in handling spatial correlations in large sensor networks. However, even graph convolutional models like STGCN and Graph WaveNet are outperformed by our proposed approach. In fact, as found in comparisons with models like GMAN, which uses an attention mechanism, their error rates are significantly lower, especially across different horizons. This shows the challenges to scale up global attention mechanisms for large graph-structured traffic systems.

To verify the reliability of the obtained performance improvements, we conduct statistical significance testing for both the average-horizon MAE and RMSE across all datasets, as depicted in Fig 5. The figure shows the relative improvement of DG-LLM over each baseline together with the corresponding significance level. We observe that all RMSE improvements are statistically significant at $p < 0.001$, while MAE improvements are statistically significant for the majority of comparisons. These results indicate that the proposed DG-LLM framework consistently reduces both average prediction error and larger error deviations relative to competing methods. Overall, these results verify that the obtained performance

**Table 5. Short-Term comparison on PeMS Dataset (PeMS04 and PeMS08).**

| Horizon | Models | PeMS04 | | | PeMS08 | | |
|---|---|---|---|---|---|---|---|
| | | MAE ↓ | RMSE ↓ | MAPE ↓ | MAE ↓ | RMSE ↓ | MAPE ↓ |
| Horizon 3 (15 mins) | STGCN | 18.552 | 29.518 | 12.74% | 15.016 | 23.346 | 9.65% |
| | GWN | 21.388 | 32.743 | 16.44% | 17.238 | 25.761 | 13.28% |
| | AGCRN | 19.320 | 32.400 | 12.81% | 15.080 | 24.000 | 9.69% |
| | TGCN | 21.840 | 35.852 | 14.67% | 17.832 | 30.106 | 13.75% |
| | GMAN | 22.855 | 42.048 | 30.59% | 17.909 | 26.656 | 13.80% |
| | ASTGCN | 19.770 | 31.200 | 13.17% | 15.750 | 24.440 | 9.70% |
| | Our Model | **18.056**±0.464 | **27.099**±0.407 | **13.07%**±1.62% | **13.768**±0.395 | **21.175**±0.535 | **10.46%**±1.51% |
| Horizon 6 (30 mins) | STGCN | 19.396 | 30.893 | 13.67% | 20.562 | 25.382 | 10.27% |
| | GWN | 20.135 | 31.658 | 16.36% | 18.726 | 25.931 | 12.53% |
| | AGCRN | 20.150 | 33.960 | 13.37% | 16.070 | 25.720 | 10.24% |
| | TGCN | 24.853 | 40.112 | 16.52% | 20.128 | 33.473 | 16.27% |
| | GMAN | 23.304 | 41.852 | 28.79% | 17.828 | 26.840 | 13.52% |
| | ASTGCN | 21.440 | 33.630 | 14.45% | 17.280 | 26.730 | 10.66% |
| | Our Model | **18.769**±0.492 | **28.214**±0.384 | **14.03%**±1.68% | **14.663**±0.286 | **22.832**±0.423 | **11.47%**±1.73% |
| Horizon 12 (60 mins) | STGCN | 21.117 | 33.223 | 15.14% | 18.437 | 28.596 | 11.74% |
| | GWN | 27.363 | 40.483 | 19.02% | 23.538 | 35.186 | 15.98% |
| | AGCRN | 21.790 | 36.860 | 14.43% | 17.670 | 28.280 | 11.23% |
| | TGCN | 24.853 | 40.112 | 16.52% | 25.061 | 40.160 | 22.99% |
| | GMAN | 31.325 | 50.917 | 32.41% | 24.945 | 36.332 | 17.63% |
| | ASTGCN | 25.350 | 39.090 | 17.44% | 20.290 | 30.930 | 12.42% |
| | Our Model | **19.969**±0.615 | **30.080**±0.513 | **14.82%**±2.09% | **15.581**±0.281 | **24.714**±0.399 | **11.56%**±1.34% |
| Average of 1-12 Horizons | STGCN | 19.489 | 30.923 | 13.67% | 16.322 | 25.427 | 10.44% |
| | GWN | 22.482 | 34.320 | 16.36% | 18.726 | 28.295 | 13.68% |
| | AGCRN | 20.230 | 34.160 | 13.44% | 16.100 | 25.760 | 10.28% |
| | TGCN | 25.505 | 40.999 | 17.24% | 20.562 | 33.920 | 17.43% |
| | GMAN | 25.297 | 44.328 | 30.41% | 19.745 | 29.251 | 14.71% |
| | ASTGCN | 21.710 | 34.120 | 14.67% | 17.360 | 26.890 | 10.69% |
| | Our Model | **18.809**±0.498 | **28.271**±0.396 | **13.84%**±1.78% | **14.507**±0.249 | **22.645**±0.364 | **10.96%**±1.33% |

improvements are not due to random variations but demonstrate the robustness and effectiveness of our proposed framework.

The error trajectory of DG-LLM remains consistently low throughout the increasing prediction horizon compared to other approaches, as shown in Fig 6. In contrast to the increasing error trajectories of all other approaches, the DG-LLM error trajectory remains relatively stable across all datasets. This demonstrates the robustness of the proposed framework in increasing the prediction horizon. It can make reliable predictions even in highly dynamic urban traffic environments.

## RQ2. Long-term forecasting

Table 6 reports long-term forecasting results on the NYC-Taxi and CH-Bike datasets. For each metric, the best-performing results are highlighted in bold, and the second-best results are underlined.

During the 48 hours, the proposed method is shown to be more effective in modeling long-range temporal dependencies than GCN- and LLM-based benchmarks. In the NYC-Taxi high-volume dataset, the proposed method achieves the lowest error across all metrics. In particular, it outperforms the LLM-based STLLM+ by decreasing the MAE to 5.4403

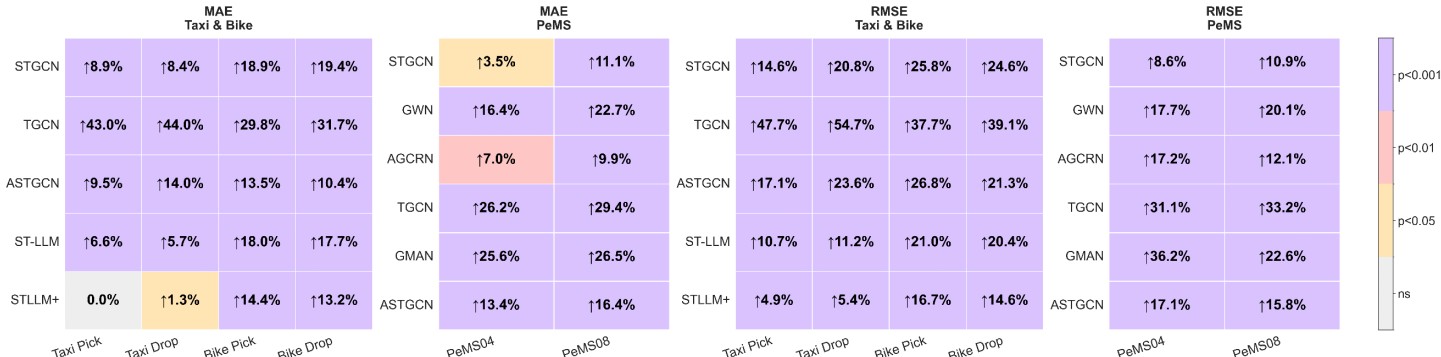

**Fig 5. Statistical significance of DG-LLM improvements.** Heatmaps show percentage reduction in MAE and RMSE relative to baseline models ($n = 5$ seeds), along with corresponding significance levels. Colors denote significance: purple ($p < 0.001$), red ($p < 0.01$), orange ($p < 0.05$), and grey (not significant). Upward arrows (↑) indicate error reduction by DG-LLM.

and the RMSE to 9.3743. Though the performance of the proposed method on the CH-Bike dataset is comparable to that of the ASTGCN method on the MAE and RMSE metrics due to the superiority of the latter on sparse data, the proposed method demonstrates superiority on the MAPE metric by reducing it to 50.16%, compared to the ST-LLM method's 57.50%. The significant gap between the proposed method and the more popular GCN-based methods, namely the TGCN and STGCN, which lack the advanced decomposition and modeling of the proposed method, further proves the effectiveness of the proposed method in reducing the error accumulation, which is common in long-range urban traffic forecasting.

## RQ3. Ablation studies

To analyze the contribution of individual components in the proposed framework, we conduct a series of ablation experiments. The results of these studies are summarized in Table 7. Starting from the full model, key modules are selectively removed while keeping all other settings unchanged. All ablation variants are trained using identical hyperparameters and evaluation protocols to ensure a fair comparison. The comparison of the impacts of these ablation studies is shown in Fig 7.

The ablation study shows that using all the suggested components is vital for achieving optimal forecasting results. Among all the modules, **Dynamic Graph Learning** has the greatest impact on forecasting results. Removing the dynamic graph module increases the MAE by approximately 8.2% on the NYC-Taxi dataset (5.0606 to 5.4773) and 16.6% on the CH-Bike dataset (1.640 to 1.9128). This demonstrates the importance of learning spatial dependencies rather than relying on the physical graph structure of roads. However, directly learning a fully dynamic graph is unstable during training. As shown in Fig 8, a curriculum learning strategy improves performance by facilitating a smooth transition from physical to learned spatial dependencies, integrating the static physical graph with the learned graph.

Furthermore, the **Spatio-temporal Encoding** module also plays a crucial role. When this module is removed, the MAE increases by 8.5% for NYC-Taxi and 16.0% for CH-Bike. This shows that the explicit encoding of spatial identity and temporal periodicity helps in learning the consistent traffic dynamics in heterogeneous regions of the city. The **VMD module** also plays a significant role in the network, as complex, non-stationary signals can be easily decomposed into simpler frequency components. When this module is removed, the MAE increases by 1.7% for NYC-Taxi and and 16.2% on CH-Bike, indicating that signal decomposition is particularly beneficial for datasets with stronger noise and variability.

Lastly, the **LoRA-based adaptation** offers an efficient way to adapt the backbone network, where removing the LoRA results in a moderate performance drop, i.e., the MAE metric increases by about 0.9% on the NYC-Taxi dataset and by 6.7% on the CH-Bike dataset, implying the effectiveness of parameter-efficient fine-tuning in adapting the pre-trained

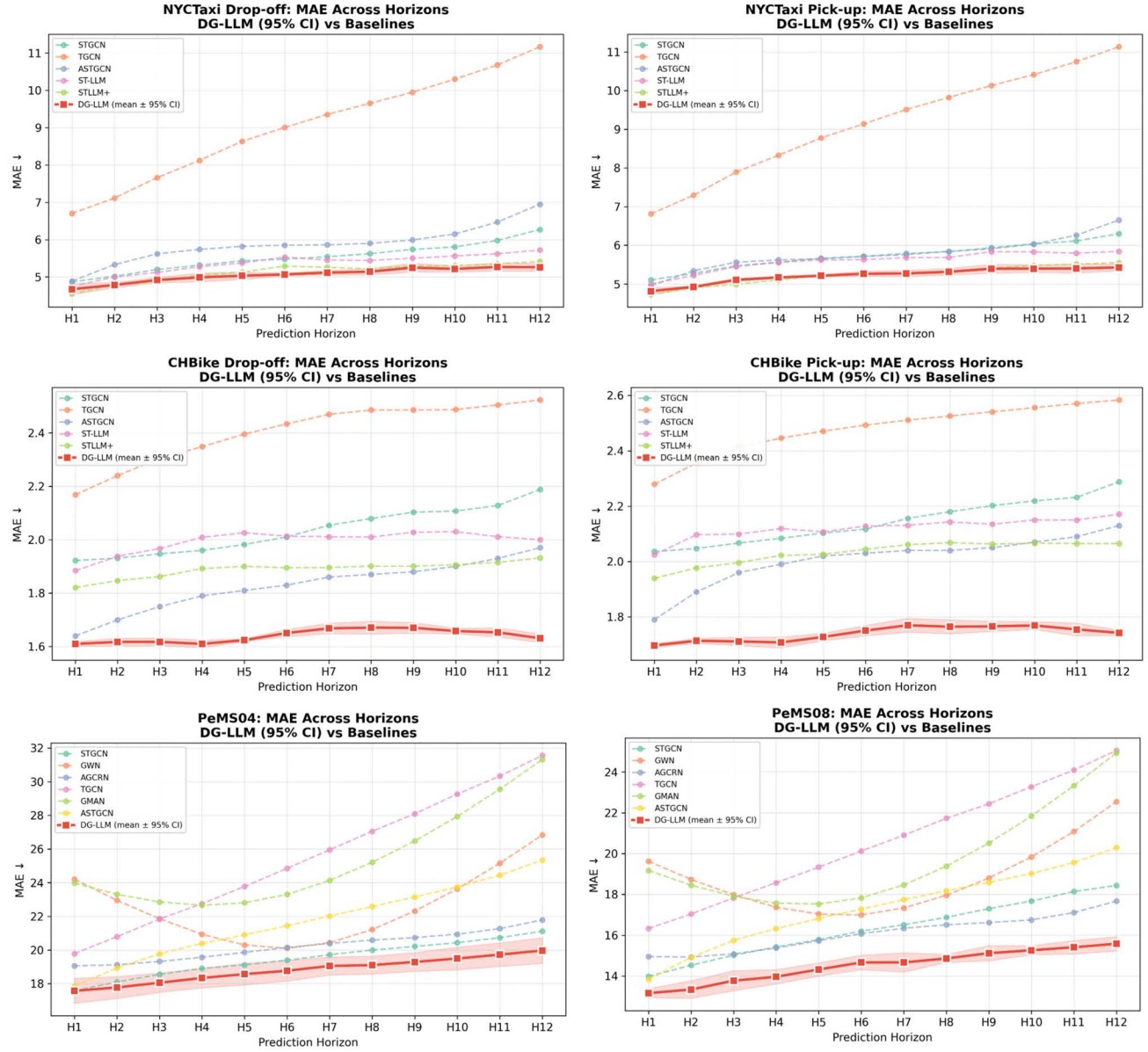

**Fig 6. Mean Absolute Error (MAE) comparison of DG-LLM against baseline models across 12 prediction horizons for all datasets.** Shaded areas represent the 95% confidence interval.

representations without overfitting. Overall, the results obtained in the present work demonstrate the potential of the proposed framework for the complementary interaction among all components, achieving robust performance on both dense taxi flow and sparse bike-sharing data.

## RQ4. Parameter analysis

To evaluate the influence of both hyperparameters and structural choices on DG-LLM, we analyze the decomposition level $K$, the number of unfrozen layers $U$, and the impact of the selected LLM backbone.

**Table 6. Long-Term Comparison of NYC-Taxi Drop-off and CH-Bike Drop-off.**

| Time | Models | NYC-Taxi Drop-off | | | CH-Bike Drop-off | | |
|------|--------|-------|-------|-------|-------|-------|-------|
| | | MAE ↓ | RMSE ↓ | MAPE ↓ | MAE ↓ | RMSE ↓ | MAPE ↓ |
| | TGCN | 8.9243 | 18.6832 | 76.16% | 2.4987 | 3.9529 | 73.20% |
| | ASTGCN | 6.4200 | 12.6500 | 44.56% | **1.9200** | 3.3200 | 60.81% |
| Horizon 96 | STGCN | 6.3806 | 14.2525 | 43.37% | 2.3090 | 3.7460 | 61.28% |
| (48 hours) | ST-LLM | 6.1932 | 11.6661 | 42.88% | 2.7169 | 4.3476 | 57.50% |
| | STLLM+ | 5.7484 | 10.5075 | 39.04% | 2.1733 | 3.3296 | 60.49% |
| | Our Model | **5.4403** | **9.3743** | **37.88%** | 2.1585 | **3.3189** | **50.16%** |

**Table 7. Ablation Study Results on NYC-Taxi and CH-Bike Drop-off Prediction.**

| Model Variant | NYC-Taxi Drop-off | | | CH-Bike Drop-off | | |
|---------------|-------|-------|-------|-------|-------|-------|
| | MAE ↓ | RMSE ↓ | MAPE ↓ | MAE ↓ | RMSE ↓ | MAPE ↓ |
| w/o VMD | 5.1477 | 9.0495 | **35.60** | 1.9063 | 2.8572 | 48.38 |
| w/o Dynamic Graph | 5.4773 | 9.4766 | 38.03 | 1.9128 | 2.8531 | 49.80 |
| w/o ST Encoding | 5.4694 | 9.4175 | 37.88 | 1.9026 | 2.8905 | 51.46 |
| w/o LoRA | 5.1075 | 8.9047 | 35.81 | 1.7509 | 2.6116 | 48.37 |
| Full Model (Ours) | **5.060**±0.043 | **8.591**±0.075 | 35.70±0.33% | **1.640**±0.009 | **2.399**±0.021 | **45.49%**±0.63% |

Each ablation variant removes a single component of the proposed framework while keeping all other settings unchanged.

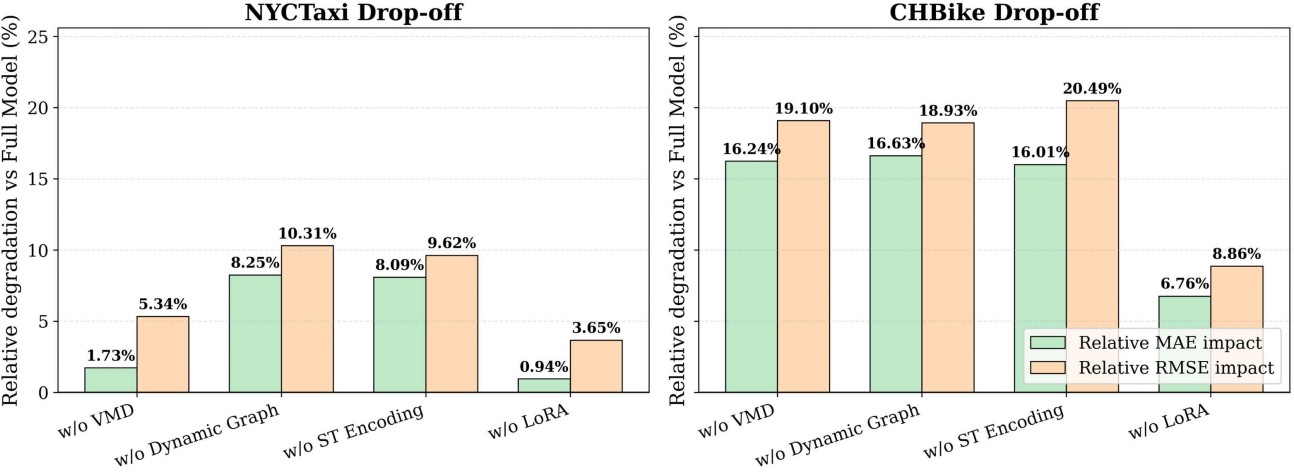

**Fig 7. Relative MAE and RMSE impact of ablation studies on the full model.**

**VMD Decomposition Level ($K$):** For the VMD decomposition level $K$, we observe that increasing $K$ from 1 to 3 on the NYC-Taxi dataset improves our model's performance as shown in Fig 9. Specifically, the MAE and RMSE values are reduced to 5.02 and 8.51, respectively, from 5.48 and 9.48. This demonstrates that moderate decomposition helps the model capture meaningful temporal patterns in the traffic signals. However, further increasing $K$ beyond 3 does not provide substantial performance benefits to our model. For example, $K = 5$ produces a lower MAE of 5.00 than $K = 3$, while increasing computational costs due to the larger number of decomposed signals. Therefore, we choose $K = 3$ as a balanced configuration as it offers a high forecasting accuracy with lower computational overhead.

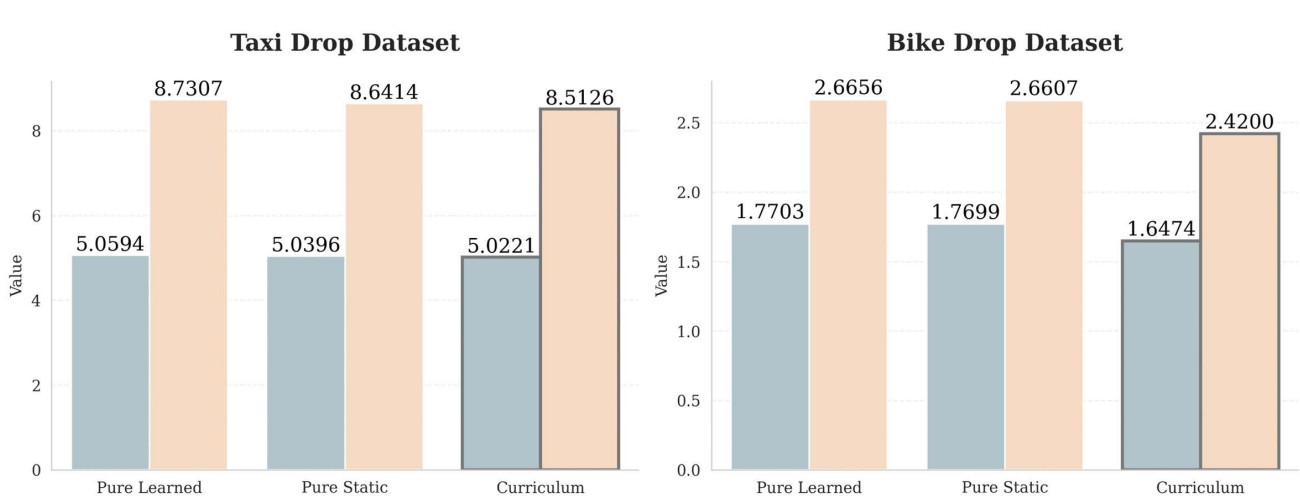

**Fig 8. MAE and RMSE comparison of curriculum learning strategy on NYC-Taxi and CH-Bike Dataset.**

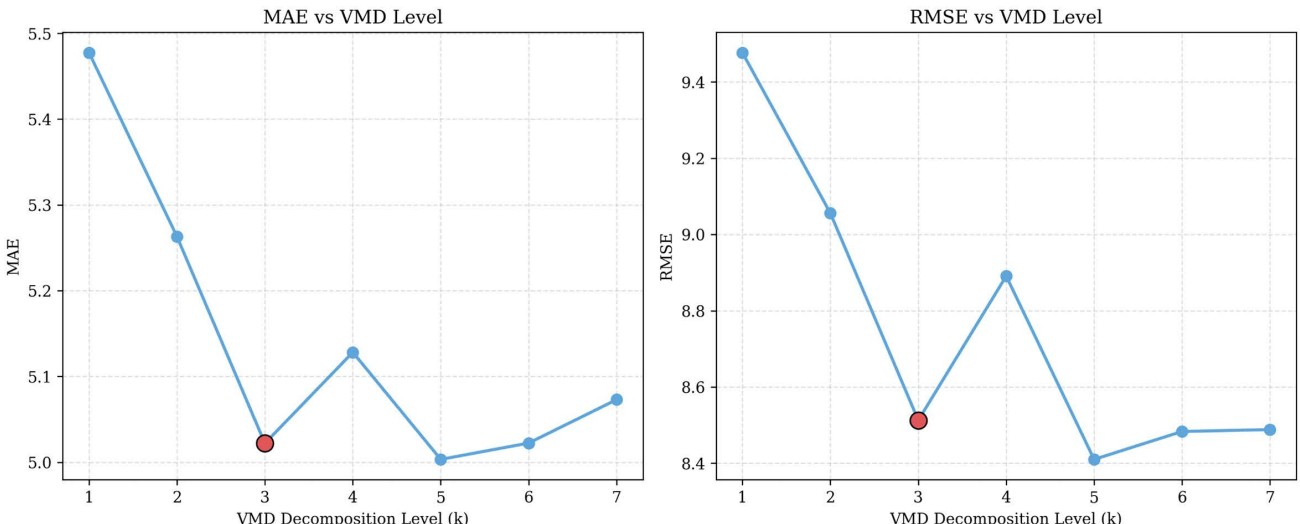

**Fig 9. VMD level comparisons on the NYC-Taxi Drop-off dataset.**

**Number of Unfrozen Layers ($U$):** From the analysis in Fig 10, the number of unfrozen LLM layers ($U$) shows different behavior for each dataset. For the NYC-Taxi dataset, the optimal performance can be seen at $U=2$ (MAE 5.10, RMSE 8.76), while further tuning $U$ from 2 to 4, the RMSE value increases to 9.14 due to overfitting. In contrast, CH-Bike needs a deeper finetuning process to adapt to its specific characteristics, with MAE and RMSE dropping steadily as $U$ moves from 1 to 3, reaching a performance peak at $U=3$ (MAE 1.74, RMSE 2.59).

**Sensitivity to Backbone Choice:** Apart from the hyperparameters, choosing an appropriate pre-trained backbone is vital for learning temporal dependencies. We compare the performances of a pre-trained GPT-2 model, a pre-trained BERT model, a pre-trained DistilGPT-2 model, a non-pretrained GPT-2 model, and a non-pretrained T5 encoder-decoder model. As illustrated

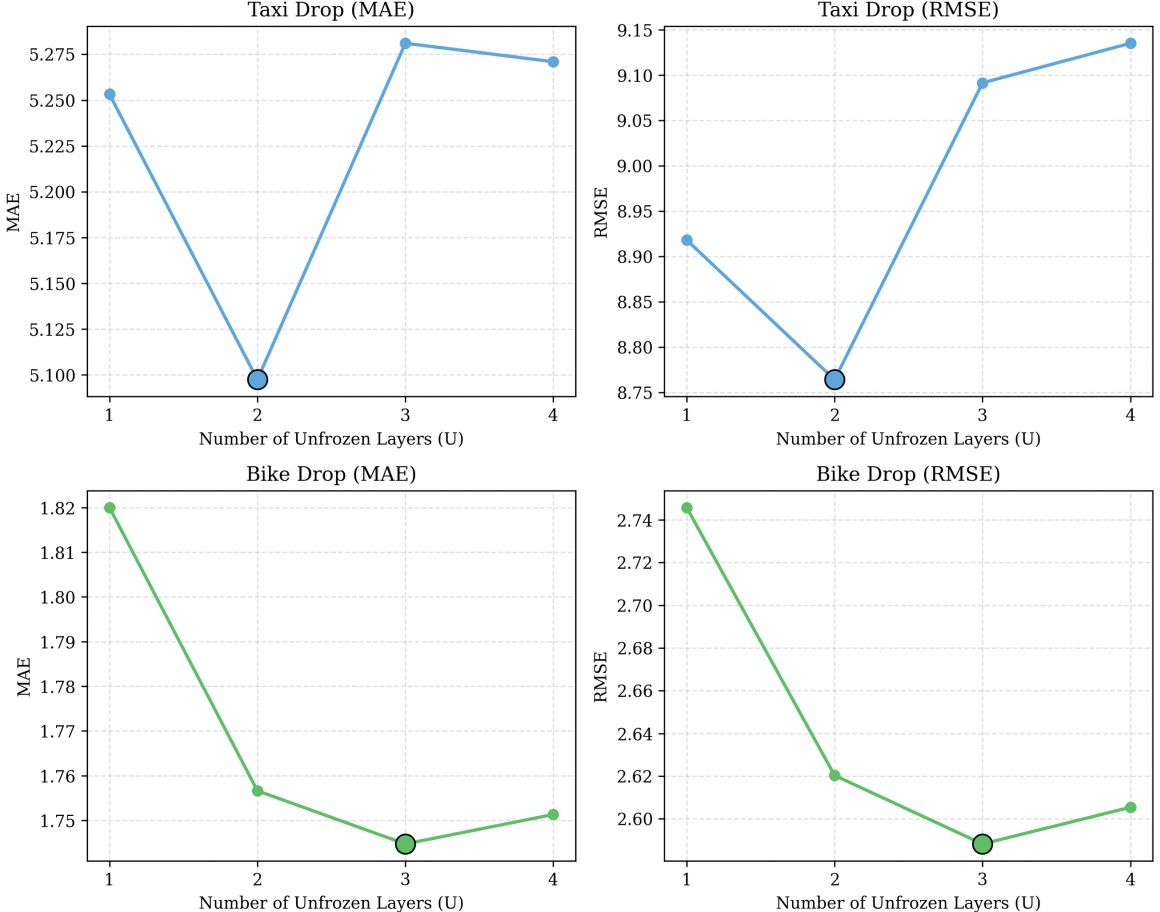

**Fig 10. MAE & RMSE Comparisons for Different Unfrozen Layers on the NYC-Taxi and CH-Bike Datasets.**

in Fig 11, the performance of pre-trained models is relatively better compared to the models without pre-training. Among all the models, the pre-trained GPT-2 models outperform others with an MAE value of 5.07 and an RMSE value of 9.20 on the NYC-Taxi dataset. On the other hand, models trained without pretraining, including the randomly initialized GPT-2, exhibit substantially higher errors, indicating that having a similar architecture alone is insufficient without pretrained knowledge. Although other pre-trained models, such as BERT and DistilGPT-2, achieve competitive results, the pre-trained GPT-2 model still performs slightly better. This is because BERT is an encoder-only model designed for bidirectional context, whereas the DistilGPT-2 model's performance is limited by its capacity. Thus, pre-trained GPT-2 remains the optimal choice for temporal dependency learning.

The above results indicate that by choosing appropriate values for $K$ and $U$, the model is able to learn complex patterns while having robust performance on various datasets. Furthermore, the superiority of the pre-trained GPT-2 backbone underscores that the choice of architecture and pre-trained knowledge is just as vital as hyperparameter tuning for achieving optimal forecasting accuracy.

## RQ5. Robustness to missing data

To evaluate the robustness in handling incomplete observations, we conducted an MCAR (Missing Completely At Random) missingness simulation by randomly masking input sequences with missing rates of 10%, 20%, 30%, and 50%. The

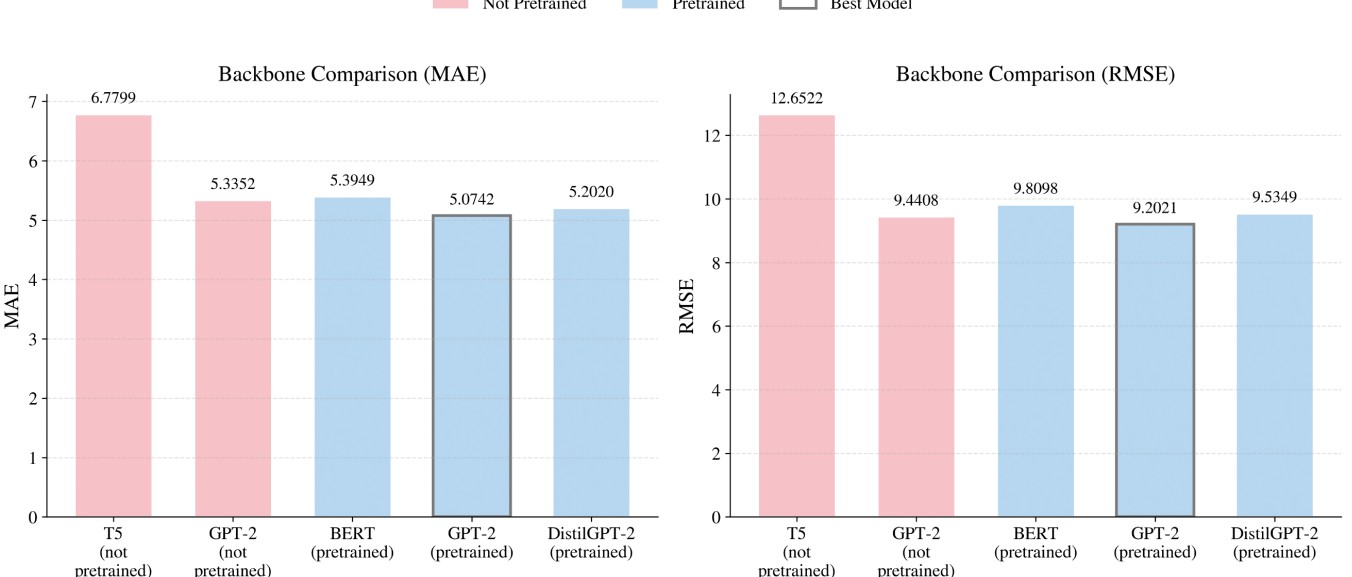

**Fig 11. MAE and RMSE Comparison on Different Pretrained and Non-Pretrained Backbones.**

experiments were conducted across both Taxi Drop and Bike Drop scenarios and compared with our base model STLLM, where DG-LLM outperforms STLLM in all missing rates in MAE and RMSE metrics, as shown in Fig 12.

The experiments demonstrated that DG-LLM performs steadily with increasing missing rates with gradual performance degradation, and maintains considerable accuracy in handling severe missing rates (50%). This further confirms that DG-LLM is more resilient to missing information and effectively exploits temporal and spatial information in handling incomplete data. In particular, for the Taxi Drop scenario, DG-LLM increases MAE from 5.0963 to 5.3216 with 10% missing information and from 5.4077 to 5.5624 with 50% missing information, while RMSE increases from 8.9431 to 9.6995 with 10% missing information and from 9.8142 to 10.4135 with 50% missing information. For Bike Drop, DG-LLM similarly increases MAE from 1.7422 to 1.9905 at 10% and from 1.8561 to 2.1172 at 50%; RMSE increases from 2.5974 to 3.0184 (10%) and from 2.8189 to 3.2852 (50%). In general, while both models perform worse with increasing missingness, DG-LLM maintains a stable margin over STLLM from moderate (30%) to severe (50%) missingness, which further justifies the robustness of our proposed design based on multi-scale temporal decomposition and dynamic graph learning.

## Computational efficiency analysis

DG-LLM adopts a parameter-efficient fine-tuning strategy, LoRA, in which only a small fraction of the model parameters are updated during training. Although the full model contains approximately 257M parameters, only about ≈7.15% (~18.4M) are trainable, while the majority of the pre-trained backbone remains frozen, as shown in Table 8. This significantly reduces the effective optimization cost compared to full fine-tuning. Furthermore, the table presents the computational efficiency analysis of the framework, including GFLOPs, training time per epoch, and inference latency across datasets. Despite adding more modules such as VMD and dynamic graph learning, DG-LLM achieves 32.7% RAM reduction over full fine-tuning while retaining moderate computational overhead and runtime efficiency. This indicates the effective balance between efficiency and computational complexity attained through the proposed architecture.

Overall, the results show that the proposed DG-LLM framework consistently achieves superior performance across diverse evaluation settings, highlighting its effectiveness for robust spatiotemporal forecasting.

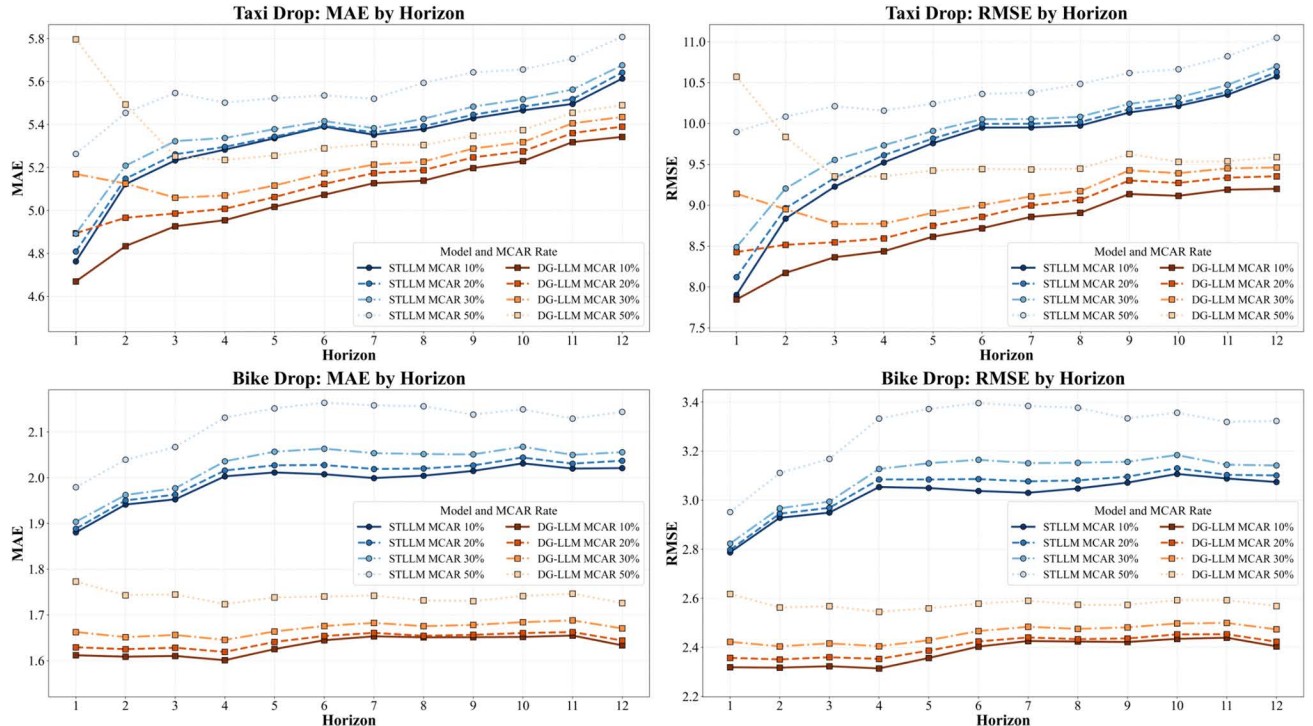

**Fig 12. MAE & RMSE Comparisons for Different Missing Rates on the NYC-Taxi and CH-Bike Datasets.**

**Table 8. Computational efficiency of the proposed framework across datasets. Parameter counts are in millions (M).**

| Dataset | Nodes | Total Params(M) | Trainable(M) | Train(%) | GFLOPs | Time/Epoch(s) | Latency(ms) |
|---|---|---|---|---|---|---|---|
| PEMSD04 | 307 | 257.81M | 18.47M | 7.17% | 93.48 | 829.8 | 292 |
| PEMSD08 | 170 | 257.70M | 18.37M | 7.13% | 49.40 | 460.0 | 183 |
| Bike Drop | 250 | 257.76M | 18.43M | 7.15% | 74.67 | 159.6 | 237 |
| Bike Pick | 250 | 257.76M | 18.43M | 7.15% | 74.67 | 165.0 | 238 |
| Taxi Drop | 266 | 257.78M | 18.44M | 7.15% | 79.89 | 176.0 | 253 |
| Taxi Pick | 266 | 257.78M | 18.44M | 7.15% | 79.89 | 181.0 | 260 |

## Discussion

To gain deeper insights into the proposed framework, we analyze its learned spatiotemporal representations, forecasting behavior, and generalization characteristics.

### Spatial graph variations

The learned adjacency matrices corresponding to the three temporal modes using the VMD algorithm are shown in Fig 13. It is evident from the figure that the relationship between the nodes depends on the frequency associated with the data. The nodes are strongly connected for low-frequency modes representing long-term and trend-based information. The patterns are caused by the general behavior of traffic dynamics in the whole network, such as daily commuting. In contrast, high-frequency modes representing sudden events, like accidents or bad weather, exhibit a sparse relationship between the nodes. The sudden changes impact a limited area of the network. Therefore, the

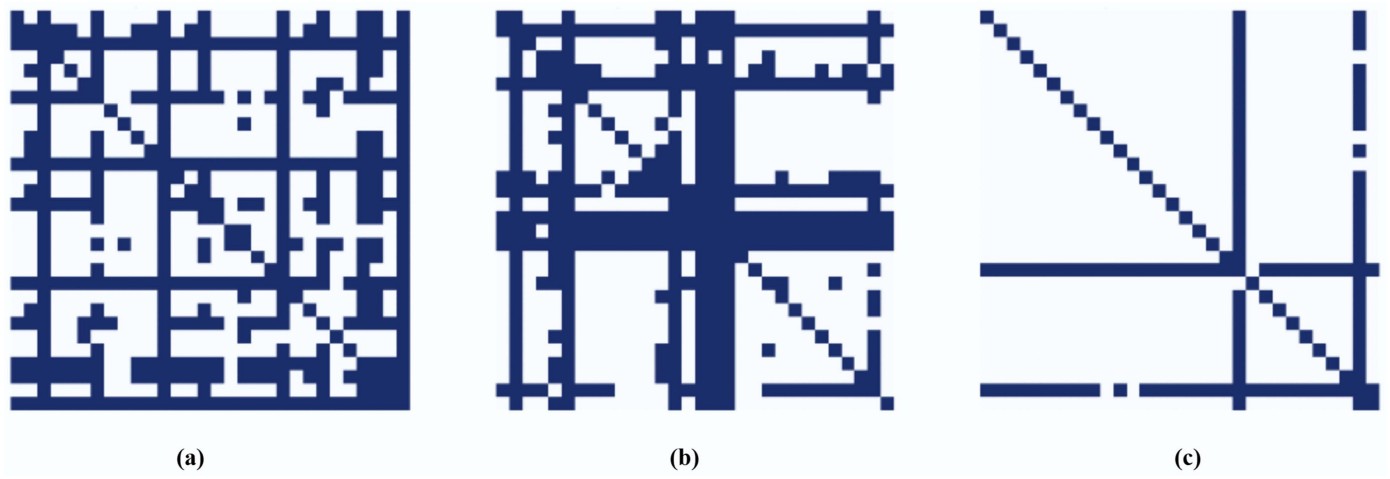

**Fig 13. Mode-dependent graphs learned from decomposed traffic signals. (a) Low Frequency, (b) Mid Frequency, and (c) High Frequency.**

nodes experience congestion only in nearby nodes. Thus, there exist variations in the dependencies of nodes on different temporal scales. Decomposing the data to isolate homogeneous components enables the model to learn distinct spatial interactions for each mode independently.

## Forecasting stability

Fig 14 shows the trajectories of short- and long-term predictions at 12- and 96-horizon forecasting horizons, respectively. While baseline methods often fail to capture sharp fluctuations during peak intervals, our model remains closely aligned with the ground truth. The stability achieved at longer horizons (e.g., 48 hours) indicates a significant reduction in cumulative error. This suggests that temporal decomposition helps preserve long-term structural patterns, while graph-aware attention ensures the model remains responsive to local temporal variations, leading to more stable long-horizon predictions.

To further demonstrate the effectiveness of the proposed model in capturing periodic patterns, we present continuous predictions across the entire test dataset for three benchmark datasets (Taxi drop-off dataset, CH-Bike drop-off dataset, and PeMS04 dataset) as shown in Fig 15. The results show that the proposed model maintains high accuracy consistently

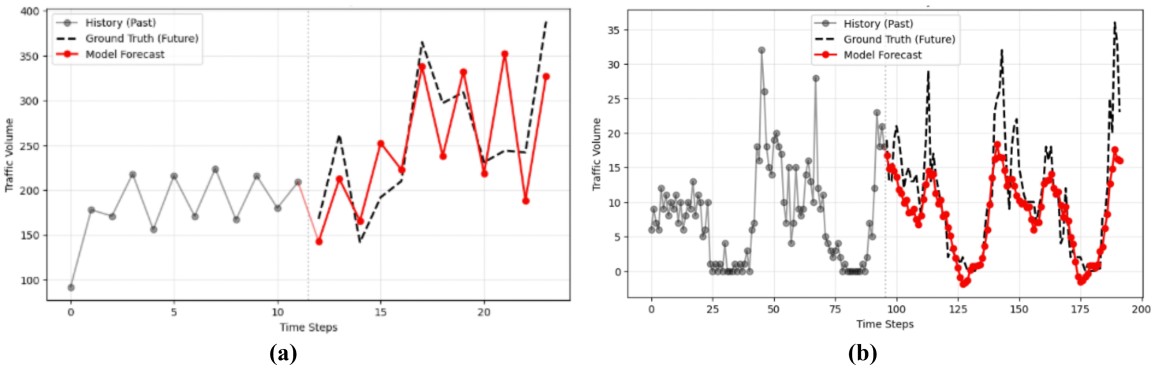

**Fig 14. Traffic forecasting visualization: (a) Short-term, (b) Long-term.**

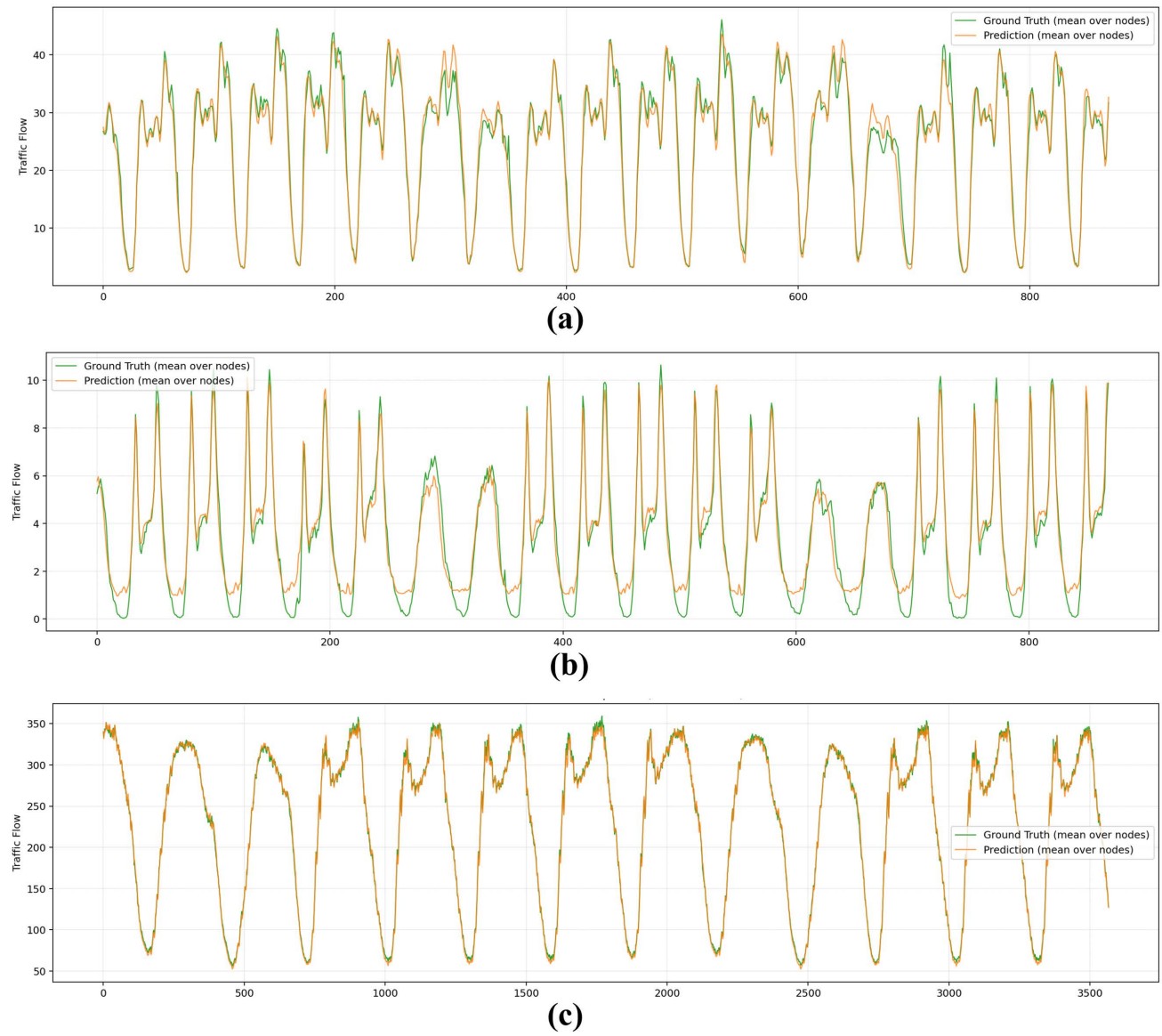

**Fig 15. Traffic forecasting results: (a)** Taxi drop-off dataset, **(b)** CH-Bike drop-off dataset, **(c)** PeMS04 dataset.

across all datasets over the full test horizon, including periodic trends such as morning and evening rush hours. This indicates that the model preserves long-term temporal structures while accurately modeling daily and weekly patterns, enabled by disentangling temporal components and reducing interference across different time scales.

## Zero-shot generalization

To assess robustness, we evaluate zero-shot transfer by training the model on one dataset and testing it on the remaining datasets without any fine-tuning. This process is repeated across all six datasets for cross-dataset evaluation. As shown in Fig 16, the model demonstrates moderate transferability across similar domains and tasks (e.g., Bike Pick-up to Bike Drop-off demand), with minimal degradation in MAE and RMSE performance. However, when we transfer from smaller-scale datasets (e.g., NYC-Taxi

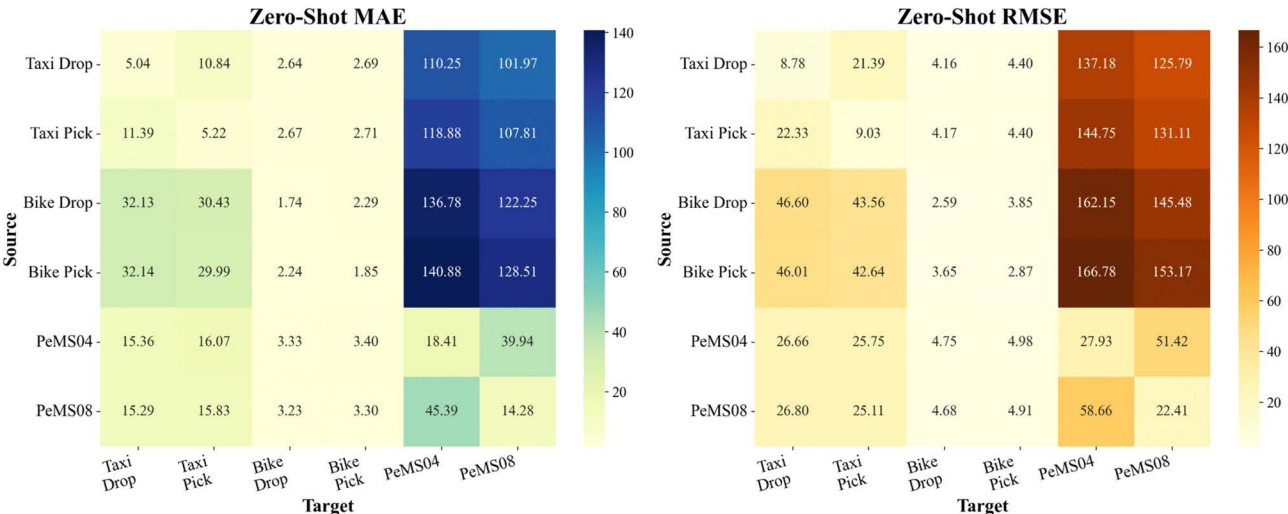

**Fig 16. MAE and RMSE for zero-shot cross-dataset transfer performance across varying urban modalities and data scales.**

and CH-Bike) to larger-scale or more complex datasets (e.g., PEMS highway data), we observe substantial performance degradation. This is due to the difficulty in generalizing from sparse data to dense data. This indicates that while the model learns transferable temporal patterns, its performance remains sensitive to differences in topology and data distribution. This suggests that the learned representations remain partially dependent on the spatial characteristics of the training data.

These findings highlight that traffic dynamics are inherently multiscale, with spatial dependencies varying across frequency components. Therefore, modeling these variations is critical for robust long-horizon forecasting.

## Conclusion

In this study, we proposed DG-LLM, a novel spatiotemporal forecasting architecture that employed decomposition-based temporal modeling and adaptive spatial relationships to represent traffic dynamics. This method disentangled traffic signals into intrinsic temporal components and learned corresponding dynamic graph representations that helped model multi-scale spatiotemporal patterns and dynamic dependencies in traffic signals. It addressed the limitations of existing methods that rely on entangled traffic signals or static spatial relationships in traffic data representations. Experimental results demonstrated that the proposed framework achieved consistent improvements over state-of-the-art methods, particularly in long-horizon forecasting scenarios where temporal instability is more pronounced. The results highlighted the importance of adaptive spatial connectivity in the representation of traffic data. Incorporating dynamic graph learning proved effective for modeling irregular road networks and heterogeneous mobility patterns. Constraining the LLM's attention mechanism through graph-aware attention masks guided the model toward spatially relevant interactions. In addition, a pretrained LLM backbone was efficiently leveraged via LoRA, reducing the fine-tuning parameter cost.

Despite its effectiveness, the proposed model still has limitations when transferred across structurally distinct traffic networks, resulting in decreased cross-domain generalization performance. This highlights the challenge of variations in topology, sensor density, and mobility patterns, which can be difficult to generalize across. Also, while parameter-efficient fine-tuning reduces training overhead, computational efficiency remains a concern due to the sequential nature of decomposition and the complexity of attention mechanisms. Future research would involve developing strategies for cross-domain generalization and transfer learning, in addition to enhancing scalability via sparse attention mechanisms and optimized input embedding for real-time deployment in large-scale urban systems.

## Supporting information

**S1 Appendix. Preliminaries.** PDF file containing the mathematical formulations and background for Graph Neural Networks (GNNs), Variational Mode Decomposition (VMD), and the Transformer architecture used in this study.
(PDF)

## Acknowledgments

The authors acknowledge the use of the Artificial Intelligence tool ChatGPT for LaTeX syntax and text refinement of the manuscript. All the content of the manuscript is reviewed and finalized by the authors. Authors take full responsibility for the content of the manuscript.

## Author contributions

**Conceptualization:** Sadia Tabassum, Naushin Nower.

**Formal analysis:** Sadia Tabassum.

**Investigation:** Sadia Tabassum.

**Methodology:** Sadia Tabassum, Naushin Nower.

**Project administration:** Naushin Nower.

**Software:** Sadia Tabassum.

**Supervision:** Naushin Nower.

**Validation:** Sadia Tabassum.

**Visualization:** Sadia Tabassum.

**Writing – original draft:** Sadia Tabassum.

**Writing – review & editing:** Naushin Nower.

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
