## [Decision Letter · Decision Letter 0]

5 Feb 2026

PONE-D-26-00024

DG-LLM: Decomposition-based dynamic graph adaptation of large language models for spatiotemporal traffic forecasting

PLOS One

Dear Dr. Nower,

Thank you for submitting your manuscript to PLOS ONE. After careful consideration, we feel that it has merit but does not fully meet PLOS ONE’s publication criteria as it currently stands. Therefore, we invite you to submit a revised version of the manuscript that addresses the points raised during the review process.

We look forward to receiving your revised manuscript.

Kind regards,

Academic Editor

PLOS One

**Journal Requirements:**

3. Please ensure that you refer to Figure 1 in your text as, if accepted, production will need this reference to link the reader to the figure.

4. Please upload a new copy of Figures 1, 2, 3, 4, 6, 7, 9, 10, 11, 12 as the detail is not clear. Please follow the link for more information:  https://journals.plos.org/plosone/s/figures

5. We note that Figure 5 in your submission contain map satellite images which may be copyrighted. All PLOS content is published under the Creative Commons Attribution License (CC BY 4.0), which means that the manuscript, images, and Supporting Information files will be freely available online, and any third party is permitted to access, download, copy, distribute, and use these materials in any way, even commercially, with proper attribution. For these reasons, we cannot publish previously copyrighted maps or satellite images created using proprietary data, such as Google software (Google Maps, Street View, and Earth). For more information, see our copyright guidelines: http://journals.plos.org/plosone/s/licenses-and-copyright.

1. You may seek permission from the original copyright holder of Figure 5 to publish the content specifically under the CC BY 4.0 license.

6. Please upload a copy of Figure 8, to which you refer in your text on page 19. If the figure is no longer to be included as part of the submission please remove all reference to it within the text.

**Additional Editor Comments:**

Note from the Editorial Office: Please be aware that Reviewer #4 and Reviewer #5 are the same reviewer. We request that you please ensure that all unique comments are addressed in full.

Reviewers' comments:

Reviewer's Responses to Questions

**Comments to the Author**

1. Is the manuscript technically sound, and do the data support the conclusions?

Reviewer #1: Yes

Reviewer #2: Yes

Reviewer #3: Yes

Reviewer #4: Yes

Reviewer #5: Yes

2. Has the statistical analysis been performed appropriately and rigorously? 

Reviewer #1: No

Reviewer #2: No

Reviewer #3: Yes

Reviewer #4: Yes

Reviewer #5: Yes

3. Have the authors made all data underlying the findings in their manuscript fully available?

Reviewer #1: Yes

Reviewer #2: Yes

Reviewer #3: Yes

Reviewer #4: Yes

Reviewer #5: Yes

4. Is the manuscript presented in an intelligible fashion and written in standard English?

Reviewer #1: Yes

Reviewer #2: Yes

Reviewer #3: Yes

Reviewer #4: Yes

Reviewer #5: Yes

5. Review Comments to the Author

Reviewer #1: The manuscript addresses an important problem in spatiotemporal traffic forecasting and proposes an interesting combination of signal decomposition, dynamic graph learning, and large language models. The experimental evaluation is extensive and conducted on multiple real-world datasets, which is a strong aspect of the work.

However, several major issues need to be addressed before the paper can be considered for publication. First, the scientific novelty is not clearly articulated. The proposed framework appears to be mainly a combination of existing techniques, and the authors should explicitly clarify what is fundamentally new in their approach.

Second, the justification for using a pretrained LLM (GPT-2) is weak. It is not convincingly demonstrated why an LLM is more suitable than standard Transformer-based time series models. A clearer motivation and, if possible, additional comparative analysis are needed.

Third, the statistical analysis lacks rigor. While MAE, RMSE, and MAPE are reported, no statistical significance tests or confidence intervals are provided, making it difficult to assess whether the reported improvements are meaningful.

Fourth, the model architecture is very complex, which raises concerns about overfitting and practical applicability. The authors should discuss the trade-off between model complexity and performance gains.

Overall, the paper has potential, but substantial revision is required to clarify the contribution, strengthen the methodology, and improve the rigor of the experimental analysis.

Reviewer #2: This manuscript presents a technically ambitious framework that combines signal decomposition, dynamic graph learning, and LLM adaptation for spatiotemporal traffic forecasting. The idea of injecting learned, mode-specific graph structures into an LLM via constrained attention is interesting, and the experimental scope is broader than many comparable studies.

The main concern is not correctness, but clarity of contribution and empirical rigor. The model is complex, with many interacting components (VMD, dynamic graphs, graph masking, LoRA, residual fusion). While the ablation study shows that removing components degrades performance, it does not clearly explain why each component is necessary or how much benefit it provides relative to its added complexity.

The statistical evaluation is another weakness. Improvements are reported without any assessment of variance or significance, which is problematic given the relatively small margins in some comparisons. Running multiple seeds and reporting confidence intervals would substantially strengthen the results.

Reproducibility is reasonably addressed through public data and code availability, but clearer documentation of preprocessing and training configurations would help.

Finally, some language in the paper overstates the contribution (e.g., claims of “bridging major gaps” or strong generalization). These should be softened to better reflect what is actually demonstrated.

Overall, the work has merit and is potentially publishable, but it requires revision to improve statistical rigor, clarify contributions, and refine presentation.

Reviewer #3: 1. Role and Necessity of the LLM

What specific capabilities of a pretrained Large Language Model are being exploited in this framework, given that the input consists of continuous traffic features rather than discrete linguistic tokens, and how does this differ from using a standard Transformer trained from scratch?

2. Justification of Using GPT-2

Why is GPT-2 chosen as the backbone LLM for traffic forecasting, and how sensitive is the proposed framework to the choice of LLM architecture (e.g., GPT-2 vs. other Transformer variants)?

3. Learning vs. Structural Bias

To what extent do the reported performance gains arise from the structural inductive biases introduced by VMD and dynamic graph masking, rather than from the pretrained knowledge embedded in the LLM itself?

4. Interpretation of “Reasoning” in LLMs

The manuscript frequently refers to the “reasoning capabilities” of LLMs; can the authors clarify what form of reasoning is expected in this numerical forecasting setting and how it is empirically demonstrated?

5. Dynamic Graph Stability and Identifiability

Since the dynamic adjacency matrix is learned directly from embeddings that themselves evolve during training, how stable and interpretable are the learned graphs across epochs and random seeds, and are multiple graph solutions equally valid?

6. Risk of Information Leakage via VMD

Although VMD is applied offline per input window, can the authors formally justify that the decomposition process does not leak future information, especially given its optimization-based formulation?

7. Fairness of Baseline Comparisons

Given that DG-LLM leverages pretrained models with millions of parameters, while several baselines are trained from scratch, how do the authors ensure a fair comparison in terms of model capacity, pretraining advantage, and computational budget?

8. Generalization Beyond Benchmarked Datasets

The evaluation is limited to six commonly used traffic datasets; how does the proposed method generalize to unseen cities, different sensor densities, or missing-sensor scenarios that are common in real-world deployments?

Reviewer #4: This manuscript introduces a novel and sophisticated framework, DG-LLM, for spatiotemporal traffic forecasting, skillfully integrating Variational Mode Decomposition (VMD), dynamic graph learning, and a pretrained LLM backbone. The work is comprehensive and demonstrates strong empirical results. However, prior to publication, major revisions are required to clarify the model's novelty against very recent literature, justify specific design choices, and provide a more critical discussion of limitations.

1. The claimed novelty regarding the integration of VMD with LLMs for traffic forecasting needs clearer positioning against the immediate predecessor, STLLM+, and other recent works. A detailed ablation study or discussion is needed to isolate the contribution of the dynamic graph versus the decomposition. Is the performance gain primarily from VMD, the dynamic graph, or their synergy?

2. The choice of GPT-2 as the LLM backbone requires more justification. Given the focus on spatial-aware adaptation, were other architectures with inherent spatial bias (e.g., Vision Transformers adapted for graphs) considered? A brief discussion on the rationale for selecting a purely temporal, causal model would be helpful.

3. The parameter analysis for VMD level K and unfrozen layers U is insightful. However, Figure 8 and Figure 9 are referenced but not included in the provided text. These critical results must be present in the final manuscript to support the conclusions about optimal hyperparameters.

4. The experiments use a fixed prediction horizon (T_out = 12) for short-term evaluation. The model's sensitivity to different horizon lengths should be analyzed. Does the advantage of VMD and the dynamic graph become more pronounced for longer-term forecasts (e.g., T_out > 12) compared to baselines?

5. While LoRA is used for efficiency, a concrete analysis of the parameter efficiency and training/inference speed compared to full fine-tuning of the LLM backbone is missing. Reporting the number of trainable parameters, training time per epoch, and inference latency would substantiate the practicality claims.

Reviewer #5: This manuscript presents an ambitious and well-structured framework (DG-LLM) for spatiotemporal traffic forecasting by integrating signal decomposition, dynamic graph learning, and large language models. The work is technically sound, the experiments are comprehensive, and the results demonstrate competitive performance. However, to solidify its contribution and ensure clarity for the broader research community, the manuscript requires revisions to sharpen its novelty claim, provide deeper methodological justification, and engage more critically with the very recent wave of literature in this rapidly advancing field.

1. The novelty of integrating VMD with an LLM needs to be more precisely demarcated from contemporaneous works. The authors rightly position their work against STLLM+, but the field is evolving quickly. The introduction and related work sections should acknowledge and differentiate the core architectural philosophy of DG-LLM (dynamic graph injection) from other emerging paradigms for spatial-aware LLMs, such as those employing vision encoders or novel attention mechanisms, to clarify its unique value proposition.

2. The dynamic graph learning module is a cornerstone of the proposal. The description of the "curriculum-inspired training strategy" (blending static and learned graphs) is a practical solution but lacks a theoretical or empirical justification. An ablation study quantifying the performance contribution of this blending strategy versus using only the learned dynamic graph would strengthen the argument for its necessity and illuminate its stabilizing effect during training.

3. The paper demonstrates strong results but would benefit from a more rigorous analysis of its generalization capabilities. The authors should consult contemporary works that address the core challenges of adapting LLMs to spatio-temporal data and domain shifts. For a comprehensive understanding, I recommend reviewing Vision-LLMs for Spatiotemporal Traffic Forecasting, arXiv; Autoregressive data generation method based on wavelet packet transform and cascaded stochastic quantization for bearing fault diagnosis under unbalanced samples, Engineering Applications of Artificial Intelligence; Multi-domain weakly decoupled domain generalization network for fault diagnosis under unknown operating conditions, Knowledge-Based Systems. These works provide insights into alternative architectural paradigms (vision-language fusion), advanced signal processing for feature learning, and robust multi-domain generalization strategies. Engaging with these concepts will help contextualize DG-LLM's approach to handling spatial data and its robustness under distribution shifts, thereby strengthening the manuscript's methodological foundations.

4. The efficiency claims related to using LoRA require substantiation. While LoRA is a known parameter-efficient fine-tuning method, the manuscript should include a concrete comparison. Reporting the total number of trainable parameters for DG-LLM versus full fine-tuning of the LLM backbone, along with comparative training times or FLOPs, would provide tangible evidence of the efficiency gains, which is crucial for assessing the framework's practicality.

5. The parameter analysis for the VMD level (K) and unfrozen layers (U) is valuable. However, the reasoning behind the selected ranges and the interpretation of the results could be deeper. For instance, why was K tested only up to 4? A discussion linking the optimal K=3 to the identifiable temporal patterns in traffic data (e.g., trend, daily periodicity, residual noise) would make the analysis more insightful.

6. The experimental design uses a fixed input and output horizon (e.g., T_in = T_out = 12). A sensitivity analysis demonstrating how the model's relative advantage over baselines changes with different prediction horizons (e.g., very short-term T_out=3 vs. long-term T_out=24) would be highly informative. It could reveal whether the benefits of decomposition and dynamic graphs are more pronounced for capturing long-range dependencies.

6. PLOS authors have the option to publish the peer review history of their article (what does this mean?). If published, this will include your full peer review and any attached files.

Reviewer #1: No

Reviewer #2: **Yes:** David Chikly

Reviewer #3: **Yes:** Muhammad Talha

Reviewer #4: No

Reviewer #5: No

---

## [Author Response · Author response to Decision Letter 1]

25 Mar 2026

Response to Reviewers’ Comments

Journal:

PLOS ONE

Manuscript ID:

PONE-D-26-00024

Title of Paper:

DG-LLM: Decomposition-based dynamic graph adaptation of large language models for spatiotemporal traffic forecasting

Authors:

Sadia Tabassum, Naushin Nower

Date Sent:

February 6, 2026

Thank you immensely for your thorough and constructive review of our manuscript. Your insightful observations have significantly contributed to refining the quality and clarity. In response to your feedback, we have diligently addressed each point, making precise modifications throughout the revised manuscript. To ensure easy identification, all revisions have been highlighted in blue within the revised manuscript. We deeply appreciate the opportunity to refine our work and are committed to addressing the reviewers' concerns effectively.

Reviewer #1:

The manuscript addresses an important problem in spatiotemporal traffic forecasting and proposes an interesting combination of signal decomposition, dynamic graph learning, and large language models. The experimental evaluation is extensive and conducted on multiple real-world datasets, which is a strong aspect of the work. However, several major issues need to be addressed before the paper can be considered for publication.

First, the scientific novelty is not clearly articulated. The proposed framework appears to be mainly a combination of existing techniques, and the authors should explicitly clarify what is fundamentally new in their approach.

Response:

We thank the reviewer for their valuable feedback. We agree that the original manuscript did not sufficiently emphasize the core scientific novelty. While our framework builds upon existing components, the novelty lies in their joint modelling to address the overlooked problem of frequency-dependent spatial heterogeneity in spatiotemporal forecasting. Specifically, we observe that spatial dependencies vary across temporal modes (e.g., trend, periodicity, and high-frequency components), a factor that is not explicitly modeled in existing approaches.

Contrary to existing models that either directly apply LLMs to the entangled time series or use static graphs, the proposed DG-LLM introduces a mode-dependent dynamic graph learning mechanism. In particular, we use Variational Mode Decomposition (VMD) to explicitly extract the intrinsic modes of the time series (i.e., trend, daily periodicity, and high-frequency noise), and show that each mode exhibits distinct spatial dependency patterns. Based on this observation, we design a dynamic graph learning module that learns separate, mode-specific graph topologies, which are then integrated into the LLM for spatiotemporal modeling.

We have revised the Introduction section (Page 3) of the manuscript to explicitly highlight the proposed modeling paradigm. The added paragraph is provided below:

DG-LLM proposes a new paradigm for spatiotemporal forecasting by simultaneously disentangling frequency-specific temporal patterns and mode-specific spatial structures before employing the LLM for spatiotemporal prediction. Unlike previous approaches that employed LLMs on raw, entangled time series data or adopted a static graph, we leverage Variational Mode Decomposition (VMD) as a task-aware pre-processing component to disentangle the time series into intrinsic modes before the LLM models temporal patterns. As a result, the proposed approach can explicitly extract the intrinsic modes of the time series (i.e., trend, daily periodicity, and high-frequency noise). We further propose a mode-dependent dynamic spatial routing mechanism that learns separate graph topologies for each time series mode, as spatial correlations are not homogeneous across frequencies; long-range trends and high-frequency noise have different dependency structures. By jointly modeling decomposed temporal patterns and their corresponding spatial structures, we capture multi-scale dependencies more effectively than existing approaches that rely on raw signals or static graph representations, thereby improving generalization across diverse datasets.

Second, the justification for using a pretrained LLM (GPT-2) is weak. It is not convincingly demonstrated why an LLM is more suitable than standard Transformer-based time series models. A clearer motivation and, if possible, additional comparative analysis are needed.

Response:

We would like to express our gratitude to the reviewer for their constructive critique. The main reason for using a pretrained LLM like GPT-2 instead of a standard Transformer model is its ability to learn rich and transferable sequence representations from large-scale pretraining. In our framework, the model operates on decomposed temporal modes combined with dynamically learned spatial structures, which introduces complex and heterogeneous patterns. Standard Transformer-based time series models, typically trained from scratch, may struggle to generalize under such conditions, especially in the presence of limited data and distribution shifts.

To validate this, we conducted a comparative analysis across multiple Transformer-based backbones, including both pretrained and non-pretrained variants. Specifically, we evaluated GPT-2 (pretrained and non-pretrained), BERT, DistilGPT-2, and T5. The result table is shown below:

Table: Comparative Analysis Across Different Pretrained and Non-Pretrained Backbones

Backbone

MAE

MAPE

RMSE

T5 (not pretrained)

6.7799

47.17%

12.6522

GPT2 (not pretrained)

5.3352

38.61%

9.4408

BERT

5.3949

36.47%

9.8098

GPT2 (our choice)

5.0742

36.17%

9.2021

DistilGPT2

5.202

36.34%

9.5349

Figure: MAE and RMSE Comparison on Different Pretrained and Non-Pretrained Backbones

The results are presented in Figure 11 (Page 22) of the revised manuscript. It shows that the pre-trained GPT-2 significantly outperforms its non-pretrained counterpart as well as other Transformer variants. This indicates that the improvements are not due to the Transformer architecture itself, but rather to the latent universal knowledge acquired during the LLM's pre-training. We have added a subsection, Sensitivity to Backbone Choice (Page 21), in the discussion section as below-

Sensitivity to Backbone Choice: Beyond numerical hyperparameters, the choice of the pre-trained backbone is critical for capturing temporal dependencies. We compare the performance of the pre-trained GPT-2 model, the pre-trained BERT model, the pre-trained DistilGPT-2 model, the GPT-2 model without pre-training, and the T5 encoder-decoder model without pre-training. As depicted in Fig. 11, the performance of the models with pre-training is comparatively better than that of the models without pre-training. Among all the models, the pre-trained GPT-2 achieves the best performance on the NYCTaxi dataset, with an MAE of 5.07 and an RMSE of 9.20. On the other hand, models trained without pretraining, including the randomly initialized GPT-2, exhibit substantially higher errors, indicating that architectural similarity alone is insufficient without pretrained knowledge. Although other pre-trained models, such as BERT and DistilGPT-2, achieve competitive results, the pre-trained GPT-2 model still performs slightly better. This is because BERT is an encoder-only model designed for bidirectional context, whereas the DistilGPT-2 model’s performance is limited by its capacity. This makes the pre-trained GPT-2 model an optimal choice for learning the temporal dependencies.

Third, the statistical analysis lacks rigor. While MAE, RMSE, and MAPE are reported, no statistical significance tests or confidence intervals are provided, making it difficult to assess whether the reported improvements are meaningful.

Response:

We thank the reviewer for highlighting the importance of rigorous statistical evaluation. We agree that the original manuscript lacked sufficient statistical analysis, and we have significantly strengthened this aspect in the revised version.

To address this issue, we have re-run the experiments for all the datasets using five different random seed values. Based on these runs, we now report the mean and 95% confidence intervals for MAE, RMSE, and MAPE as shown below:

Table: Multiseed Results with 95% interval for all Datasets

Dataset

MAE Mean

MAE

95% CI

RMSE Mean

RMSE

95% CI

MAPE(%) Mean

MAPE(%) 95% CI

Taxi Drop

5.0594

[5.006, 5.113]

8.5906

[8.498, 8.684]

35.7

[35.28, 36.11]

Taxi Pick

5.2234

[5.180, 5.267]

8.7743

[8.693, 8.856]

34.76

[33.51, 36.00]

Bike Drop

1.6403

[1.630, 1.651]

2.3993

[2.373, 2.425]

45.49

[44.71, 46.26]

Bike Pick

1.7393

[1.726, 1.753]

2.6491

[2.615, 2.683]

48.17

[47.17, 49.17]

PEMSD04

18.8093

[18.19, 19.43]

28.2706

[27.78, 28.76]

13.84

[11.63, 16.05]

PEMSD08

14.5071

[14.20, 14.82]

22.6447

[22.19, 23.10]

10.96

[9.31, 12.60]

In addition, to assess whether the observed improvements are statistically significant, we conducted paired t-tests on the MAE and RMSE values obtained from DG-LLM and the baseline models across all six datasets. The results show that the improvements of DG-LLM are statistically significant (p < 0.001) across most datasets. The statistical accuracy values are provided in Figure 5 (Page 19) as given below:

Figure: Statistical significance of DG-LLM improvements.Heatmaps show percentage reduction in MAE and RMSE relative to baseline models (n=5 seeds), along with corresponding significance levels. Colors denote significance: purple (p < 0.001), red (p < 0.01), orange (p < 0.05), and grey (not significant). Upward arrows (↑) indicate error reduction by DG-LLM.

We addressed this in our revised manuscript in the Results section (Pages 18-19) as below-

To verify the reliability of the obtained performance improvements, we conduct statistical significance testing for both the average-horizon MAE and RMSE across all datasets, as depicted in Fig. 5. The figure shows the relative improvement of DG-LLM over each baseline together with the corresponding significance level. We observe that all RMSE improvements are statistically significant at p<0.001, while MAE improvements are statistically significant for the majority of comparisons. These results indicate that the proposed DG-LLM framework consistently reduces both average prediction error and larger error deviations relative to competing methods. Overall, these results verify that the obtained performance improvements are not due to random variations but demonstrate the robustness and effectiveness of our proposed framework.

Fourth, the model architecture is very complex, which raises concerns about overfitting and practical applicability. The authors should discuss the trade-off between model complexity and performance gains.

Response:

We thank the reviewer for this important comment. While DG-LLM introduces additional components (e.g., VMD decomposition and dynamic graph learning), these modules are designed to better capture the complex spatio-temporal patterns in traffic data. The VMD layer helps separate stable temporal patterns from high-frequency noise, allowing the model to learn from more structured and less entangled signals. This reduces the burden on the model and mitigates overfitting by separating stable patterns from high-frequency noise.

In addition, the DG-LLM model utilizes a parameter-efficient fine-tuning method (LoRA), in which the majority of the layers of the pre-trained LLM are kept frozen during the fine-tuning process. As shown in Table 9, only 7% of the parameters of the pre-trained model are fine-tunable, while more than 92% of the parameters are frozen, which limits the capability of the pre-trained model during the fine-tuning process. In terms of the trade-off between complexity and performance, we observe that each component of DG-LLM contributes to measurable performance gains, as demonstrated in our ablation studies (Page 20). At the same time, the parameter-efficient design ensures that the additional architectural complexity is moderate. As reported in the revised manuscript, DG-LLM maintains moderate computational complexity (in terms of GFLOPs) and achieves substantial memory savings (approximately 32.7% reduction in RAM usage compared to full fine-tuning). We have added a new subsection, “Computational Efficiency Analysis,” on page 22, as given below.

DG-LLM adopts a parameter-efficient fine-tuning strategy, LoRA, in which only a small fraction of the model parameters are updated during training. Although the full model contains approximately 257M parameters, only about 7.15% (~18.4M) are trainable, while the majority of the pre-trained backbone remains frozen as shown in Table 8. This significantly reduces the effective optimization cost compared to full fine-tuning. In addition, Table 8 reports the computational efficiency across datasets, including GFLOPs, training time per epoch, and inference latency. Despite incorporating additional components such as VMD and dynamic graph learning, DG-LLM reduces RAM usage by 32.7% compared to full fine-tuning while maintaining moderate computational complexity and practical runtime performance. These results demonstrate that the proposed framework achieves a favorable balance between model capacity and efficiency for large-scale spatiotemporal forecasting.

Table: Computational efficiency of DG-LLM across datasets. Parameter counts are in millions (M).

Dataset

Nodes

Total Params (M)

Trainable (M)

Train %

GFLOPs

Time/

Epoch (s)

Latency (ms)

Taxi Drop

266

257.78

18.44

7.15%

79.89

176

253

Taxi Pick

266

257.78

18.44

7.15%

79.89

181

260

Bike Drop

250

257.76

18.43

7.15%

74.67

159.6

237

Bike Pick

250

257.76

18.43

7.15%

74.67

165

238

PEMSD04

307

257.81

18.47

7.17%

93.48

829.8

292

PEMSD08

170

257.7

18.37

7.13%

49.4

460

183

Overall, the paper has potential, but substantial revision is required to clarify the contribution, strengthen the methodology, and improve the rigor of the experimental analysis.

Reviewer #2:

This manuscript presents a technically ambitious framework that combines signal decomposition, dynamic graph learning, and LLM adaptation for spatiotemporal traffic forecasting. The idea of injecting learned, mode-specific graph structures into an LLM via constrained attention is interesting, and the experimental scope is broader than many comparable studies.

The main concern is not correctness, but clarity of contribution and empirical rigor. The model is complex, with many interacting components (VMD, dynamic graphs, graph masking, LoRA, residual fusion). While the ablation study shows that removing components degrades performance, it does not clearly explain why each component is necessary or how much benefit it provides relative to its added complexity.

Response:

We would like to express our sincere gratitude to the reviewer for their positive feedback and insightful criticism. We agree that it is important not only to show performance degradation in ablation studies, but also to clearly explain the role of each component and justify the added architectural complexity.

To address this, we have extended the ablation analysis in the revised manuscript by explicitly quantifying and interpreting the contribution of each component. As shown in Figure 7 (Page 20), different modules have varying levels of impact on performance. In particular, removing the dynamic graph learning module results in the largest degradation (up to ~10% on NYCTaxi and ~20% on CHBike), highlighting its critical role in modeling mode-dependent spatial relationships. The spatiotemporal encoding module also shows a substantial impact, while VMD contributes by reducing temporal entanglement and improving learning stability. The LoRA module has comparatively smaller performance gain, but it improves efficiency with minimal added complexity.

Figure: Relative MAE and RMSE impact of ablation studies on the full model

These results demonstrate that each component addresses a specific limitation (e.g., temporal entanglement, static spatial assumptions, or inefficient fine-tuning) and that the performance gains are concentrated in key modules rather than uniformly distributed.

We have incorporated this detailed analysis into the ablation study discussion in the Resul

---

## [Decision Letter · Decision Letter 1]

14 Apr 2026

PONE-D-26-00024R1

DG-LLM: Decomposition-based dynamic graph adaptation of large language models for spatiotemporal traffic forecasting

PLOS One

Dear Dr. Nower,

Thank you for submitting your manuscript to PLOS ONE. After careful consideration, we feel that it has merit but does not fully meet PLOS ONE’s publication criteria as it currently stands. Therefore, we invite you to submit a revised version of the manuscript that addresses the points raised during the review process.

We look forward to receiving your revised manuscript.

Kind regards,

Shih-Lin Lin, Ph.D

Academic Editor

PLOS One

Journal Requirements:

Reviewer's Responses to Questions

**Comments to the Author**

1. If the authors have adequately addressed your comments raised in a previous round of review and you feel that this manuscript is now acceptable for publication, you may indicate that here to bypass the “Comments to the Author” section, enter your conflict of interest statement in the “Confidential to Editor” section, and submit your "Accept" recommendation.

Reviewer #1: All comments have been addressed

Reviewer #6: (No Response)

2. Is the manuscript technically sound, and do the data support the conclusions?

Reviewer #1: Yes

Reviewer #6: (No Response)

3. Has the statistical analysis been performed appropriately and rigorously?

Reviewer #1: (No Response)

Reviewer #6: (No Response)

4. Have the authors made all data underlying the findings in their manuscript fully available?

Reviewer #1: Yes

Reviewer #6: (No Response)

5. Is the manuscript presented in an intelligible fashion and written in standard English?

Reviewer #1: Yes

Reviewer #6: (No Response)

6. Review Comments to the Author

Reviewer #1: The authors have adequately addressed the concerns raised in the previous round of review, and the manuscript has improved significantly as a result.

The contribution is now clearly presented, and the justification of the proposed approach has been strengthened. The addition of statistical analysis, including multi-seed experiments, confidence intervals, and significance testing, improves the reliability of the results.

The manuscript is well structured and written in clear and understandable English. The experimental evaluation is comprehensive and supports the conclusions.

Recommendation: Accept.

Reviewer #6: This study proposes DG-LLM, a spatiotemporal traffic forecasting framework that combines signal decomposition, dynamic graph learning, and pretrained large language models with efficient fine-tuning to capture multi-scale temporal patterns and spatial dependencies across diverse traffic datasets. It requires a major revision based on:

1. The abstract is overly dense with methodological components (e.g., decomposition, dynamic graphs, LLM integration, LoRA), which obscures the core novelty; the authors should streamline the description and clearly emphasize the primary contribution and innovation.

2. Although relative improvements in MAE and RMSE are reported, the abstract lacks baseline references, dataset scale, and evaluation protocol details; providing this context is necessary to properly assess the significance and robustness of the reported gains

3. The introduction should clearly conclude with a distinct section highlighting the novel contributions of your work.

4. At the ending of the intro, it is advised to add a para that mentions briefly what each next section contains.

5. The literature review should benefit from more explorations of previous studies.

6. The discussion section needs to be expanded to more thoroughly analyze the results.

7. The first paragraph of the conclusion should succinctly summarize the contributions of the study in past tense.

8. The second paragraph of the conclusion should provide clear and actionable future recommendations.

9. Equations are not properly cited, please add original references.

10. Please include the response letter separately with different colors.

7. PLOS authors have the option to publish the peer review history of their article (what does this mean?). If published, this will include your full peer review and any attached files.

Reviewer #1: No

Reviewer #6: No

---

## [Author Response · Author response to Decision Letter 2]

21 Apr 2026

Response to Reviewers’ Comments

Journal: PLOS ONE

Manuscript ID: PONE-D-26-00024R1

Title of Paper: DG-LLM: Decomposition-based dynamic graph adaptation of large language models for spatiotemporal traffic forecasting

Authors:Sadia Tabassum, Naushin Nower

Date Sent: April 15, 2026

We sincerely thank the reviewer for the thorough and constructive evaluation of our manuscript. The insightful comments and suggestions have been invaluable in refining the clarity, quality, and overall presentation of our work. In response, we have addressed all the points and made corresponding revisions throughout the manuscript. For ease of reference, all modifications have been highlighted in blue in the revised manuscript. We greatly appreciate having an opportunity to improve the quality of our manuscript through your constructive suggestions.

Reviewer #6:

1. The abstract is overly dense with methodological components (e.g., decomposition, dynamic graphs, LLM integration, LoRA), which obscures the core novelty; the authors should streamline the description and clearly emphasize the primary contribution and innovation.

Response:

Thank you for this valuable observation. We have revised the abstract to improve readability by simplifying the description and focusing more clearly on the primary contribution and innovation of the proposed DG-LLM framework (Page 1). The revised part of the abstract paragraph is provided below:

… In this paper, we present a novel framework named DG-LLM that leverages the advantages of decomposed temporal representations and adaptive spatial connectivity to model spatiotemporal dependencies. In this framework, traffic signals are decomposed into intrinsic modes, and dynamic graphs are learned for each mode to represent the spatial dependencies. These representations are then incorporated with pre-trained Large Language Models for effective long-range temporal dependency modeling.

2. Although relative improvements in MAE and RMSE are reported, the abstract lacks baseline references, dataset scale, and evaluation protocol details; providing this context is necessary to properly assess the significance and robustness of the reported gains

Response:

We appreciate this suggestion. In the revised abstract, we have incorporated additional contextual information, including references to baseline models, the datasets used, and a brief indication of the evaluation setting (Page 1). This provides a clearer understanding of the significance and robustness of the reported improvements. The revised part of the abstract paragraph is provided below:

… We conducted comprehensive experiments across six real-world traffic datasets spanning urban mobility systems and highway traffic networks and evaluated short- and long-term forecasting. Experimental results demonstrate that our framework provides significant improvements over state-of-the-art approaches, including benchmark graph- and LLM-based spatiotemporal forecasting models, even in long-term forecasting scenarios with severe temporal instability. Our model outperforms other methods by achieving 13-19% improvements in MAE and 19-25% in RMSE across all six benchmarks compared with baseline approaches. Additional analyses, including ablation studies, robustness to missing data, and zero-shot cross-dataset evaluation, further validate the effectiveness and generalization capability of the proposed framework.

3. The introduction should clearly conclude with a distinct section highlighting the novel contributions of your work.

Response:

Thank you for the comment. We have revised the final part of the Introduction to more clearly and explicitly present the novel contributions of this work (Pages 3-4). The contributions are now described in a concise and structured manner to improve clarity. The revised paragraph is provided below:

The major contributions of this work are summarized as follows:

In this paper, we propose a decomposition-based spatiotemporal forecasting framework that models multi-scale temporal patterns and their corresponding spatial dependencies, thereby addressing the limitations of existing methods that operate on entangled time series.

Within this framework, we design a mode-dependent dynamic graph learning mechanism that constructs a separate graph for each decomposed temporal mode to capture the homogeneous patterns, enabling adaptive modeling of traffic interactions across different frequency components.

We further integrate these mode-specific graphs into a graph-aware pretrained large language model, enabling it to leverage dynamic spatial structures while effectively capturing long-range temporal dependencies.

We validate the effectiveness of the proposed framework through a comprehensive experimental evaluation across various benchmark real-world datasets, showing significant improvements in both short- and long-term forecasting scenarios.

4. At the ending of the intro, it is advised to add a para that mentions briefly what each next section contains.

Response:

We agree that this improves readability. A new paragraph has been added at the end of the Introduction (Page 4), briefly describing the organization of the manuscript and the contents of each section. The added paragraph is provided below:

The remainder of this paper is organized as follows. The Related Work section discusses previous works on graph-based, LLM-based, and decomposition-based approaches. The Problem Statement section introduces problem formulation and notations. The Methodology section presents the proposed DG-LLM framework. In the Experiments Section, we discuss our dataset, baselines, evaluation metrics, and experimental setup. The Results section presents the experimental results for both short-term and long-term forecasting, along with ablation studies and additional analyses. The Discussion section presents the findings, and the Conclusion section concludes the paper, outlining future research directions.

5. The literature review should benefit from more explorations of previous studies.

Response:

We appreciate the reviewer’s valuable feedback. The Related Work section has been substantially revised and expanded to provide a more comprehensive exploration of prior studies (Pages 4-6).

The revised section now includes a structured discussion of the evolution of traffic forecasting methods, ranging from classical statistical models (e.g., ARIMA, Kalman filtering) to modern deep learning paradigms, including graph-, LLM-, and decomposition-based architectures, highlighting their respective strengths and limitations. We have also incorporated additional relevant works on traffic forecasting and LLM-based approaches to broaden the literature review.

We further refined the analysis within each category by emphasizing key methodological developments and existing limitations. The research gap has also been more clearly articulated by contrasting graph-based, LLM-based, and decomposition-based approaches, emphasizing their limitations in jointly modeling multi-scale traffic patterns and dynamic spatial dependencies.

6. The discussion section needs to be expanded to more thoroughly analyze the results.

Response:

We sincerely appreciate the reviewer's valuable comment. In our revised paper, the Discussion section has been significantly improved, with further analysis of the experimental findings (Pages 23-24). We added a deeper interpretation of the learned mode-dependent graph structures, explained the complementary contribution of the components to the observed forecasting gains, and discussed dataset-specific behavior across urban demand and highway traffic settings. We also strengthened the analysis of long-horizon forecasting stability and zero-shot generalization, and explicitly highlighted the limitations and transferability challenges of the proposed framework. The revised Discussion section is provided below.

To gain deeper insights into the proposed framework, we analyze its learned spatiotemporal representations, forecasting behavior, and generalization characteristics.

Spatial Graph Variations: The learned adjacency matrices corresponding to the three temporal modes using the VMD algorithm are shown in Fig. 13. It is evident from the figure that the relationship between the nodes depends on the frequency associated with the data. The nodes are strongly connected for low-frequency modes representing long-term and trend-based information. The patterns are caused by the general behavior of traffic dynamics in the whole network, such as daily commuting. In contrast, high-frequency modes representing sudden events, like accidents or bad weather, exhibit a sparse relationship between the nodes. The sudden changes impact a limited area of the network. Therefore, the nodes experience congestion only in nearby nodes. Thus, there exist variations in the dependencies of nodes on different temporal scales. Decomposing the data to isolate homogeneous components enables the model to learn distinct spatial interactions for each mode independently.

Fig 13: Mode-dependent graphs learned from decomposed traffic signals. (a) Low Frequency, (b) Mid Frequency, and (c) High Frequency

Forecasting Stability: Fig. 14 shows the trajectories of short- and long-term predictions at 12- and 96-horizon forecasting horizons, respectively. While baseline methods often fail to capture sharp fluctuations during peak intervals, our model remains closely aligned with the ground truth. The stability achieved at longer horizons (e.g., 48 hours) indicates a significant reduction in cumulative error. This suggests that temporal decomposition helps preserve long-term structural patterns, while graph-aware attention ensures the model remains responsive to local temporal variations, leading to more stable long-horizon predictions.

Fig 14: Traffic forecasting visualization: (a) Short-term, (b) Long-term.

To further demonstrate the effectiveness of the proposed model in capturing periodic patterns, we present continuous predictions across the entire test dataset for three benchmark datasets (Taxi drop-off dataset, CH-Bike drop-off dataset, and PeMS04 dataset) as shown in Fig 15. The results show that the proposed model maintains high accuracy consistently across all datasets over the full test horizon, including periodic trends such as morning and evening rush hours. This indicates that the model preserves long-term temporal structures while accurately modeling daily and weekly patterns, enabled by disentangling temporal components and reducing interference across different time scales.

Fig 15: Traffic forecasting results: (a) Taxi drop-off dataset, (b) CH-Bike drop-off dataset, (c) PeMS04 dataset.

Zero-Shot Generalization: To assess robustness, we evaluate zero-shot transfer by training the model on one dataset and testing it on the remaining datasets without any fine-tuning. This process is repeated across all six datasets for cross-dataset evaluation. As shown in Fig. 16, the model demonstrates moderate transferability across similar domains and tasks (e.g., Bike Pick-up to Bike Drop-off demand), with minimal degradation in MAE and RMSE performance. However, when we transfer from smaller-scale datasets (e.g., NYC-Taxi and CH-Bike) to larger-scale or more complex datasets (e.g., PEMS highway data), we observe substantial performance degradation. This is due to the difficulty in generalizing from sparse data to dense data. This indicates that while the model learns transferable temporal patterns, its performance remains sensitive to differences in topology and data distribution. This suggests that the learned representations remain partially dependent on the spatial characteristics of the training data.

Fig 16: MAE and RMSE for zero-shot cross-dataset transfer performance across varying urban modalities and data scales.

These findings highlight that traffic dynamics are inherently multiscale, with spatial dependencies varying across frequency components. Therefore, modeling these variations is critical for robust long-horizon forecasting.

7. The first paragraph of the conclusion should succinctly summarize the contributions of the study in past tense.

Response:

Thank you for the suggestion. The first paragraph of the Conclusion has been revised to provide a concise summary of the study in the past tense, clearly reflecting the study's contributions and outcomes (Page 25). The revised paragraph is provided below:

In this study, we proposed DG-LLM, a novel spatiotemporal forecasting architecture that employed decomposition-based temporal modeling and adaptive spatial relationships to represent traffic dynamics. This method disentangled traffic signals into intrinsic temporal components and learned corresponding dynamic graph representations that helped model multi-scale spatiotemporal patterns and dynamic dependencies in traffic signals. It addressed the limitations of existing methods that rely on entangled traffic signals or static spatial relationships in traffic data representations. Experimental results demonstrated that the proposed framework achieved consistent improvements over state-of-the-art methods, particularly in long-horizon forecasting scenarios where temporal instability is more pronounced. The results highlighted the importance of adaptive spatial connectivity in the representation of traffic data. Incorporating dynamic graph learning proved effective for modeling irregular road networks and heterogeneous mobility patterns. Constraining the LLM’s attention mechanism through graph-aware attention masks guided the model toward spatially relevant interactions. In addition, a pretrained LLM backbone was efficiently leveraged via LoRA, reducing the fine-tuning parameter cost.

8. The second paragraph of the conclusion should provide clear and actionable future recommendations.

Response:

We have revised the second paragraph of the Conclusion to include clear, actionable directions for future research (Page 25). These revisions provide a more forward-looking perspective on potential extensions of this work. The revised paragraph is provided below:

Despite its effectiveness, the proposed model still has limitations when transferred across structurally distinct traffic networks, resulting in decreased cross-domain generalization performance. This highlights the challenge of variations in topology, sensor density, and mobility patterns, which can be difficult to generalize across. Also, while parameter-efficient fine-tuning reduces training overhead, computational efficiency remains a concern due to the sequential nature of decomposition and the complexity of attention mechanisms. Future research would involve developing strategies for cross-domain generalization and transfer learning, in addition to enhancing scalability via sparse attention mechanisms and optimized input embedding for real-time deployment in large-scale urban systems.

9. Equations are not properly cited, please add original references.

Response:

We appreciate this valuable comment. In the revised manuscript, we have carefully reviewed all equations and added appropriate citations to their original sources (e.g., VMD, attention mechanism, graph attention, and LoRA). In addition, we have a dedicated appendix (S1 Appendix: Preliminaries) that provides a more detailed formulation and discussion of these foundational methods with proper attribution (Pages 26-28).

10. Please include the response letter separately with different colors.

Response:

We thank the reviewer for this suggestion. The response letter has been prepared separately in accordance with PLOS One guidelines, with distinct color formatting used to clearly differentiate reviewer comments and author responses.

---

## [Decision Letter · Decision Letter 2]

30 Apr 2026

DG-LLM: Decomposition-based dynamic graph adaptation of large language models for spatiotemporal traffic forecasting

PONE-D-26-00024R2

Dear Dr. Nower,

We’re pleased to inform you that your manuscript has been judged scientifically suitable for publication and will be formally accepted for publication once it meets all outstanding technical requirements.

Kind regards,

Shih-Lin Lin, Ph.D

Academic Editor

PLOS One

Additional Editor Comments (optional):

Reviewers' comments:

Reviewer's Responses to Questions

**Comments to the Author**

1. If the authors have adequately addressed your comments raised in a previous round of review and you feel that this manuscript is now acceptable for publication, you may indicate that here to bypass the “Comments to the Author” section, enter your conflict of interest statement in the “Confidential to Editor” section, and submit your "Accept" recommendation.

Reviewer #6: (No Response)

2. Is the manuscript technically sound, and do the data support the conclusions?

Reviewer #6: (No Response)

3. Has the statistical analysis been performed appropriately and rigorously? 

Reviewer #6: (No Response)

4. Have the authors made all data underlying the findings in their manuscript fully available?

Reviewer #6: (No Response)

5. Is the manuscript presented in an intelligible fashion and written in standard English?

Reviewer #6: (No Response)

6. Review Comments to the Author

Reviewer #6: (No Response)

7. PLOS authors have the option to publish the peer review history of their article (what does this mean?). If published, this will include your full peer review and any attached files.

Reviewer #6: No

---

## [Editor Report · Acceptance letter]

PONE-D-26-00024R2

PLOS One

Dear Dr. Nower,

I'm pleased to inform you that your manuscript has been deemed suitable for publication in PLOS One. Congratulations! Your manuscript is now being handed over to our production team.

Kind regards,

on behalf of

Professor Shih-Lin Lin

Academic Editor

PLOS One